# ON THE SPECIALIZATION OF NEURAL MODULES

**Devon Jarvis**[1,2,*]**, Richard Klein**[1]**, Benjamin Rosman**[1,3] **& Andrew M. Saxe**[2,3]
[1]School of Computer Science and Applied Mathematics, University of the Witwatersrand
[2]Gatsby Computational Neuroscience Unit & Sainsbury Wellcome Centre, UCL
[3]CIFAR Azrieli Global Scholar, CIFAR
{devon.jarvis,richard.klein,benjamin.rosman1}@wits.ac.za
a.saxe@ucl.ac.uk

## ABSTRACT

A number of machine learning models have been proposed with the goal of achieving systematic generalization: the ability to reason about new situations by combining aspects of previous experiences. These models leverage compositional architectures which aim to learn specialized modules dedicated to structures in a task that can be composed to solve novel problems with similar structures. While the compositionality of these architectures is guaranteed by design, the modules specializing is not. Here we theoretically study the ability of network modules to specialize to useful structures in a dataset and achieve systematic generalization. To this end we introduce a minimal space of datasets motivated by practical systematic generalization benchmarks. From this space of datasets we present a mathematical definition of systematicity and study the learning dynamics of linear neural modules when solving components of the task. Our results shed light on the difficulty of module specialization, what is required for modules to successfully specialize, and the necessity of modular architectures to achieve systematicity. Finally, we confirm that the theoretical results in our tractable setting generalize to more complex datasets and non-linear architectures.

## 1 INTRODUCTION

Humans frequently display the ability to *systematically generalize*, that is, to leverage specific learning experiences in diverse new settings (Lake et al., 2019). For instance, exploiting the approximate compositionality of natural language, humans can combine a finite set of words or phonemes into a near-infinite set of sentences, words, and meanings. Someone who understands "brown dog" and "black cat" also likely understands "brown cat," to take one example from Szabó (2012). The result is that a human's ability to reason about situations or phenomena extends far beyond their ability to directly experience and learn from all such situations or phenomena.

Deep learning techniques have made great strides in tasks like machine translation and language prediction, providing proof of principle that they can succeed in quasi-compositional domains. However, these methods are typically data hungry and the same networks often fail to generalize in even simple settings when training data are scarce (Lake & Baroni, 2018b; Lake et al., 2019). Empirically, the degree of systematicity in deep networks is influenced by many factors. One possibility is that the learning dynamics in a deep network could impart an implicit inductive bias toward systematic structure (Hupkes et al., 2020); however, a number of studies have identified situations where depth alone is insufficient for structured generalization (Pollack, 1990; Niklasson & Sharkey, 1992; Phillips & Wiles, 1993; Lake & Baroni, 2018b; Mittal et al., 2022). Another significant factor is architectural modularity, which can enable a system to generalize when modules are appropriately configured (Vani et al., 2021; Phillips, 1995). However, identifying the right modularity through learning remains challenging (Mittal et al., 2022). In spite of these (and many other) possibilities for improving systematicity (Hupkes et al., 2020), it remains unclear when standard deep neural networks will exhibit systematic generalization (Dankers et al., 2021), reflecting a long-standing theoretical debate stretching back to the first wave of connectionist deep networks (Rumelhart & McClelland, 1986; Pollack, 1990; Fodor & Pylyshyn, 1988; Smolensky, 1991; 1990; Hadley, 1993; 1994).

In this work we theoretically study the ability of neural modules to specialize to structures in a dataset. Our goal is to provide a formalism for systematic generalization and to begin to concretize some of

---

*Corresponding author

the intuitions and concepts in the systematic generalization literature. To begin we make a careful distinction between the compositionality and systematicity of a neural network architecture. Specifically, in this work we maintain that compositionality is a feature of an architecture, such as a Neural Module Network (Andreas et al., 2016; Andreas, 2018), or dataset where modules or components can be *composed* by design. Systematicity is a property of a (potentially compositional) architecture exploiting structure in the world (dataset) such as the compositional structure of natural language from the Szabó (2012) example. Intuitively, if a dataset does not have structure which can be exploited for generalization then no compositional architecture will be able to systematically generalize. As a result in this work we are concerned with formalizing **both** the dataset and neural network learning dynamics to study this interplay between domain and architecture. The main approach of this work can be summarized as follows: we introduce a reflective space of datasets that contain compositional and non-compositional features, and examine the impact of implicit biases and architectural modularity on the learned input-output mappings of deep linear network modules. In particular,

- We derive exact training dynamics for deep linear network modules as a function of the dataset parameters. This is a novel, theoretical means of analysing the effect of dataset structure on neural network learning dynamics.
- We formalize the goal of modularity as finding lower-rank sub-structure within a dataset that can be exploited to improve generalization.
- We show that for all datasets in the space, despite the possibility of learning a systematic mapping, non-modular networks do not do so under gradient descent dynamics.
- We show that modular network architectures can learn fully systematic network mappings, but only when the modularity perfectly segregates the underlying lower-rank sub-structure in the dataset.

In Section 7 we show that our findings, which rely on a simplified setting for mathematical tractability, generalize to more complicated datasets and non-linear architectures by training a convolutional neural network to label handwritten digits between 0 and 999. Overall, our results help clarify the interplay between dataset structure and architectural biases which can facilitate systematic generalization when neural modules specialize.

## 2 BACKGROUND

Systematic generalization has been proposed as a key feature of intelligent learning agents which can generalize to novel stimuli in their environment (Hockett & Hockett, 1960; Fodor & Pylyshyn, 1988; Hadley, 1993; Kirby et al., 2015; Lake et al., 2017; Mittal et al., 2022). In particular, the closely related concept of compositional structure has been shown to have benefits for both learning speed (Shalev-Shwartz & Shashua, 2016; Ren et al., 2019) and generalizability (Lazaridou et al., 2018). There are, however, counter-examples which find only a weak correlation between compositionality and generalization (Andreas, 2018) or learning speed (Kharitonov & Baroni, 2020). In most cases neural networks do not manage to generalize systematically (Ruis et al., 2020; Mittal et al., 2022), or systematic generalization occurs only with the addition of explicit regularizers or a degree of supervision on the learned features (Shalev-Shwartz et al., 2017; Wies et al., 2022) which is also termed "mediated perception" (Shalev-Shwartz & Shashua, 2016).

Neural Module Networks (NMNs) (Andreas et al., 2016; Hu et al., 2017; 2018) have become one successful method of creating network architectures which generalize systematically. By (jointly) training individual neural modules on particular subsets of data or to perform particular subtasks, the modules will specialize. These modules can then be combined in new ways when an unseen data point is input to the model. Thus, through the composition of the modules, the model will systematically generalize, assuming that the correct modules can be structured together. Bahdanau et al. (2019b) show, however, that strong regularizers are required for the correct module structures to be learned. Specifically, it was shown that neural modules become coupled and as a result do not specialize in a manner which can be useful to the compositional design of the architecture. Thus, without task-specific regularizers, systematic mappings did not emerge with NMNs.

Similar problems arise with other models which are compositional in nature. Tensor Product Networks (Smolensky et al., 2022) for example aim to learn an encoding of place and content for features in a data point. These encodings are then tensor-produced and summed. By the nature of separating

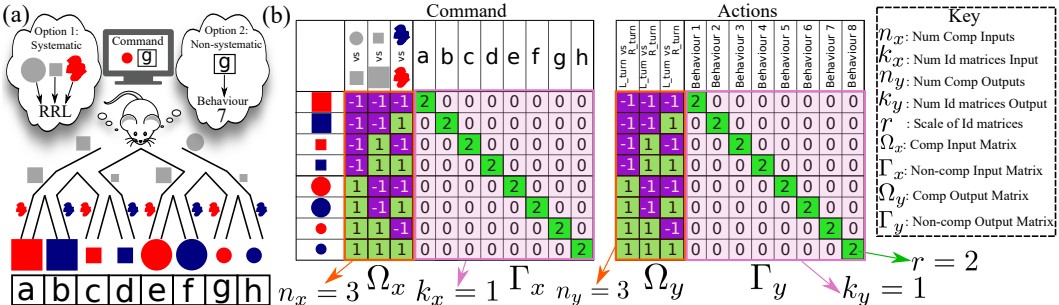

Figure 1: Problem setting and dataset space. (a) When navigating a maze towards a target, an agent might extract various input features, mapping these to a sequence of actions. (b) We schematize this setting with a space of datasets containing compositional ($\Omega$) and non-compositional ($\Gamma$) features in the input (middle panel) and output (right panel). Rows contain examples and columns contain features. In this case objects are identified with both a compositional component (based on features: size, shape and colour) and non-compositional component (based on absolute position).

"what" and "where" it is possible to systematically generalize when seen objects appear in unseen positions. For example, in language a noun which has only been seen as the subject in a sentence can be easily understood as the object of a novel sentence as the word meaning is separate from its position. These models, however, require coherent encodings to be learned for similar unseen objects which is conceptually similar to requiring neural modules to specialize and not guaranteed. Similarly, Recurrent Independent Mechanisms (RIMs) (Goyal et al., 2019) and Compositional Recursive Learners (CRLs) (Chang et al., 2018) experience difficulties with the network modules becoming coupled and only worked consistently when paired with the same modules that were used for the training data. Thus, a number of questions remain unanswered. What is required for neural modules to specialize? Is there a natural tendency for neural modules to specialize in the absence of strong regularizers? How much does the dataset structure matter for systematic generalization? These are all questions we aim to help address in this work. To this end we now define the space of datasets used in our analysis and provide a definition of systematicity for this space.

## 3   A SPACE OF DATASETS WITH COMPOSITIONAL SUB-STRUCTURE

The notion of systematic generalization is broad, and has been assessed using a variety of datasets and paradigms (Hupkes et al., 2020). Here we introduce a simple setting motivated by the SCAN (Lake & Baroni, 2018a) and grounded-SCAN (gSCAN) datasets (Ruis et al., 2020) commonly used to evaluate systematic generalization with practical neural models. Other similar benchmarks have also been used in prior work (Hupkes et al., 2019; Bahdanau et al., 2019a). In the SCAN and gSCAN datasets a command is given to an agent such as "jump left twice" or "walk to the red small circle cautiously". In the case of gSCAN an image of a grid-world is also presented depicting the agent and a number of objects sharing similar features. These features include small vs large, red vs blue vs green vs yellow, circle vs square vs rhombus. The agent is then tasked with producing a sequence of primitive actions such as "L_turn", "R_turn", "WALK". In the case where an agent is told to carry out an action in a specific manner (an adverb is used in the command) a sequence of memorized actions must also be completed. For example to perform an action "cautiously" means that the action string "L_turn, R_turn, R_turn, L_turn" must be output before each step forward.

For our space of datasets we aimed to imitate the primary features of these established benchmarks while making sure the task remains linearly solvable to allow for our theoretical analysis. An example of one dataset from the space of datasets is depicted in Fig. 1. Conceptually the task is for an agent to navigate a series of paths to reach the desired object in the domain as specified by a command. The command which is input to the model specifies the features of the object to be found where the features are "large vs small", "square vs circle", "red vs blue". The agent is also told the index of the object in the 1-dimensional space of objects and so it also knows the absolute position in the environment. The agent must then output a string of three actions where each action is either "L_turn" or "R_turn". Finally, in the agent's output we also include a set of unique actions which correspond to the agent executing an entire memorized behaviour (one per object in the environment). These full-behaviour outputs are reminiscent of the agent in gSCAN remembering that the "cautiously" command maps to a sequence of four actions (looking left and right).

To formalize this setting, we define a parametric space of datasets with input and output matrices $X = [\Omega_x \, \Gamma_x]^T$ and $Y = [\Omega_y \, \Gamma_y]^T$ respectively, where $n_x, n_y, k_x, k_y, r \in \mathbb{Z}^+$ are the parameters that define a specific dataset. Figure 1(b) visualizes one of the simpler datasets in the space. The *compositional input feature* matrix $\Omega_x \in \{-1, 1\}^{2^{n_x} \times n_x}$ consists of all binary patterns with $n_x$ bits. Here $n_x$ is a key parameter determining the number of bits in the compositional input structure. Overall, the dataset contains $2^{n_x}$ examples. The *compositional output feature* matrix $\Omega_y \in \{-1, 1\}^{2^{n_x} \times n_y}$ is a uniform sampling (although any sampling method would work) of $n_y$ features (columns) from $\Omega_x$, and is the compositional component of the output matrix. By using only a sampling of $\Omega_x$ features for $\Omega_y$ we account for the case where some visually distinct features are not necessary for navigation. Next, the *non-compositional input feature* matrix $\Gamma_x = [rI_1 \, ... \, rI_{k_x}]$ consists of $k_x$ scaled identity matrices, $I_i \in \{0, 1\}^{2^{n_x} \times 2^{n_x}}$. Similarly, the *non-compositional output* matrix $\Gamma_y = [rI_1 \, ... \, rI_{k_y}]$ has $k_y$ scaled identity matrices $I_i \in \{0, 1\}^{2^{n_x} \times 2^{n_x}}$ with scale factor $r$. These identity matrices provide a single feature for each pattern which is only on for that pattern. This space of datasets is consistent with numerical notions of compositionality in previous works (Andreas, 2018), which define compositionality as a homomorphism between the observation space and the naming space (since the input-output mappings are linear they are homomorphic). Crucially for our analysis, the amount of compositional structure can be titrated by adjusting $n_x$ and $n_y$. Similarly, $k_x, k_y$ and $r$ control the frequency and intensity of the non-compositional features, which are both factors that can promote non-compositional language being used by humans (Rogers et al., 2004).

## 3.1 Systematicity as Exploiting Lower-rank Sub-structure

Novel to our analysis, datasets in this space allow redundant solutions: the compositional output component can be generated based on compositional input features alone (systematic mappings), but they can equally be generated using non-compositional features alone, or some mixture of the two (non-systematic mappings). This formalization of the datasets is motivated by common systematicity benchmarks, but it is also reflective of a more general fact: that in many settings there are multiple ways to solve a problem. However, it is not the case that all approaches generalize equally well. This means that the inductive biases placed upon a model which influence the kinds of mappings it learns is a key consideration, even if the difference is not apparent on training data. In Sections 5 and 6 we aim to formalize and study the architectural inductive biases which push a model towards systematic mappings. We begin in this section with a mathematical definition of systematicity for our space of datasets.

While some prior theoretical works have defined systematic generalization, such as the grading from weak to semantic systematicity (Hadley, 1994; Bodén & Niklasson, 2000), these works have remained behavioural and relied on linguistic notions of syntax. Thus, a general mathematical definition of systematicity has remained elusive. In more recent empirical studies (Bahdanau et al., 2019b; Lake & Baroni, 2018b) it is intuitive based on the domain what systematicity would be. However, this context dependent nature of systematicity is the root of the difficulty in defining it. Thus, Definition 3.1 offers a formal definition of systematicity for our space of datasets.

**Definition 3.1.** Systematic generalization is the identification and learning of lower-rank sub-structure $(\Omega_x, \Omega_y)$ in the full dataset $(X, Y)$ such that the rank of the population covariance on the sub-structure: $E[\Omega_x \Omega_x^T], E[\Omega_y \Omega_x^T]$ is lower than the rank of the population covariance on the dataset as a whole: $E[XX^T], E[YX^T]$. Consequently, it is more probable that a training sample will be full rank on the sub-structure than on the full dataset, which facilitates generalization on the sub-structure.

The definition above relies on the rank of the covariance matrix to define systematicity. It is important then to note that by partitioning portions of the feature or output space it is possible to change the rank of the sub-problems which emerge. Take for example the dataset shown in Figure 1. The rank of the entire dataset covariance matrix is $8$ due to the identity block being orthogonal. Thus, for the network to learn the full mapping it must see all $8$ data points. However, if we only consider the compositional inputs and outputs the rank of this portion of the covariance matrix (the compositional covariance) is $3$. Thus, even though there are $8$ unique data points a model would only need to see $3$ data points which are not linearly independent to learn that portion of the mapping if learned in isolation. For 3 binary variables the probability of obtaining a full-rank compositional covariance matrix from 3 samples is $57\%$ and from 4 samples is $91\%$ (see Appendix B for the calculation of the sample probabilities for 3 and 4 compositional features). Thus, if the model were to learn the mapping between compositional sub-structures in isolation it would be nearly certain to generalize to the entire dataset having seen only half of the data points in the training set. This is our notion of systematicity: the ability to learn portions of a mapping with lower-rank sub-structure distinct from

the rest of the mapping which increases the rank of the problem. The benefit of this is that the network will generalize far better on the structured portions of the input or output space (see Appendix A for a motivating example of this point and for more intuition on Definition 3.1). This benefit also grows with the number of compositional features since adding more compositional features results in an exponential increase in the number of data points which can be generalized to, relative to the number training examples needed (shown in Appendix B). With the space of datasets defined, we now describe the primary theoretical tool in this work. That is to calculate the training dynamics of both shallow and deep linear networks for all datasets in our space.

# 4 LEARNING DYNAMICS IN SHALLOW AND DEEP LINEAR NETWORKS

While deep linear networks can only represent linear input-output mappings, the dynamics of learning change dramatically with the introduction of one or more hidden layers (Fukumizu, 1998; Saxe et al., 2014; 2019; Arora et al., 2018; Lampinen & Ganguli, 2019), and the learning problem becomes non-convex (Baldi & Hornik, 1989). They therefore serve as a tractable model of the influence of depth specifically on learning dynamics, which prior work has shown to impart a low-rank inductive bias on the linear mapping (Huh et al., 2021).

We leverage known exact solutions to the dynamics of learning from small random weights in deep linear networks (Saxe et al., 2014; 2019) to describe the full learning trajectory analytically for every dataset in our space. We take the novel theoretical approach of writing these dynamics in terms of the dataset parameters for our space of datasets (Equations 4-9 and 11-12). This allows us to analyse the effect of dataset structure on the training dynamics and learned mappings. In particular, consider a single hidden layer network computing output $\hat{y} = W^2 W^1 x$ in response to an input $x$, trained to minimize the mean squared error loss using full batch gradient descent with small learning rate $\epsilon$ (full details and technical assumptions are given in Appendix C). The network's total input-output map after $t$ epochs of training is

$$W^2(t)W^1(t) = UA(t)V^T, \tag{1}$$

where $A(t)$ is a diagonal matrix of singular values. The dynamics of $A(t)$, as well as the orthogonal matrices $U$ and $V$, depend on the singular value decomposition of the input- and input-output correlations in the dataset. If the input- and input-output correlations can be expressed as

$$\Sigma^x = E[XX^T] = VDV^T, \quad \Sigma^{yx} = E[YX^T] = USV^T \tag{2}$$

where $U$ and $V$ are orthogonal matrices of singular vectors and $S, D$ are diagonal matrices of singular values/eigenvalues, then the diagonal elements of $A(t)_{\alpha\alpha} = \pi_\alpha(t)$ evolve through time as

$$\pi_\alpha(t) = \frac{\lambda_\alpha/\delta_\alpha}{1 - \left(1 - \frac{\lambda_\alpha}{\delta_\alpha \pi_0}\right) \exp\left(\frac{-2\lambda_\alpha}{\tau} t\right)}, \tag{3}$$

where $\lambda_\alpha$ and $\delta_\alpha$ are the associated input-output singular value and input eigenvalue ($S_{\alpha\alpha}$ and $D_{\alpha\alpha}$ respectively), $\pi_0$ denotes the singular value at initialization, and $\tau = \frac{1}{2^{n_x}\epsilon}$ is the learning time constant. These dynamics describe a trajectory which begins at the initial value $\pi_0$ when $t = 0$ and increases to the correct asymptotic value $\pi_\alpha^* = \lambda_\alpha/\delta_\alpha$ as $t \to \infty$. This trajectory corresponds to the network learning the covariance between the input and output data - the correct mapping.

In essence, the network's total input-output mapping at all times in training is a function of the singular value decomposition of the dataset statistics. We find that there are three distinct input-output singular values which we denote $\lambda_1$, $\lambda_2$ and $\lambda_3$; two distinct input singular values $\delta_1$ and $\delta_2$; and therefore three asymptotes $\pi_1^*, \pi_2^*$ and $\pi_3^*$ (the final point for each SV trajectory $\pi_1, \pi_2$ and $\pi_3$):

$$\lambda_1 = \left(\frac{(k_x r^2 + 2^{n_x})(k_y r^2 + 2^{n_x})}{2^{2n_x}}\right)^{\frac{1}{2}} \tag{4}$$

$$\pi_1^* = \lambda_1/\delta_1 = \left(\frac{k_y r^2 + 2^{n_x}}{k_x r^2 + 2^{n_x}}\right)^{\frac{1}{2}} \tag{5}$$

$$\lambda_2 = \left(\frac{(k_x r^2 + 2^{n_x})(k_y r^2)}{2^{2n_x}}\right)^{\frac{1}{2}} \tag{6}$$

$$\pi_2^* = \lambda_2/\delta_1 = \left(\frac{k_y r^2}{k_x r^2 + 2^{n_x}}\right)^{\frac{1}{2}} \tag{7}$$

$$\lambda_3 = \left(\frac{k_x k_y r^4}{2^{2n_x}}\right)^{\frac{1}{2}} \tag{8}$$

$$\pi_3^* = \lambda_3/\delta_2 = \left(\frac{k_y}{k_x}\right)^{\frac{1}{2}} \tag{9}$$

By writing this decomposition in terms of the dataset parameters, substituting these expressions into the dynamics equation (Equation 3) gives equations for the networks mapping and full learning trajectories at all times during training for all datasets in the space. Full derivations, including explicit expressions for singular vectors, are deferred to Appendix D.

A similar derivation for a shallow network (no hidden layer) shows that the singular values of the model's mapping follow the trajectory

$$\pi_\alpha(t) = \lambda_\alpha/\delta_\alpha \left(1 - \exp\left(-\delta_\alpha t/\tau\right)\right) + \pi_0 \exp\left(-\delta_\alpha t/\tau\right), \tag{10}$$

such that the time course depends on the singular values of the input covariance matrix, $\Sigma^x$. The unique singular values are

$$\delta_1 = \frac{(k_x r^2 + 2^{n_x})}{2^{n_x}} \tag{11}$$

$$\delta_2 = \frac{k_x r^2}{2^{n_x}}. \tag{12}$$

To empirically verify our theoretical results in Equations 4 to 9 we simulate the full training dynamics for deep and shallow linear networks trained using gradient descent on an instantiation from the space of datasets in Figure 2. While training, we compute the singular values of the network after each epoch of training. These simulations of the training dynamics for each unique singular value are then compared to the predicted dynamics. We see close agreement between the predicted and simulated trajectories.[1] In the following sections we utilise these equations for the training dynamics to analyse to what extent neural networks naturally specialise and display systematicity.

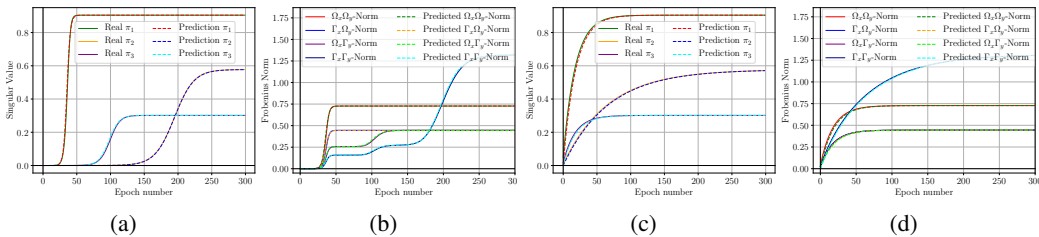

|  (a)  |  (b)  |  (c)  |  (d)  |

Figure 2: Unlike shallow networks, deep networks show distinct stages of improvement over learning. (Panels a-d): Analytical learning dynamics for deep (a,b) and shallow (c,d) linear networks. (a,c) Comparisons of predicted (dotted) and actual (solid) singular value trajectories over learning, for one dataset's singular values. (b,d) Comparisons of predicted (dotted) and actual (solid) Frobenius norms of the input-output mapping to/from compositional $(\Omega_x, \Omega_y)$ and non-compositional $(\Gamma_x, \Gamma_y)$ features. *Parameters*: $n_x = 3, k_x = 3, n_y = 1, k_y = 1, r = 1$.

## 5    THE EVOLUTION OF SYSTEMATICITY OVER LEARNING

With the decomposition of the training dynamics and a definition of systematicity for our space of datasets in hand we now look to understand the extent to which a network comes to rely on lower-rank sub-structure in a dataset to generalize systematically. To do this we calculate the Frobenius norm of the network's mapping (the function implemented by the network transforming inputs to outputs) between different subsets of input and output components. To account for all datasets in the space and obtain the full dynamics over training, the norms are expressed in terms of the modes of variation. In particular we split the input-output mapping into four partitions: compositional inputs $(\Omega_x)$ to compositional outputs $(\Omega_y)$; non-compositional inputs $(\Gamma_x)$ to compositional outputs $(\Omega_y)$; compositional inputs $(\Omega_x)$ to non-compositional outputs $(\Gamma_y)$; and non-compositional inputs $(\Gamma_x)$ to non-compositional outputs $(\Gamma_y)$. These norms provide a precise measure of the network's association between input and output blocks. For example, $\Gamma_x\Omega_y$-Norm depicts how much the network relies on the non-compositional input component $(\Gamma_x)$ to label the compositional output component $(\Omega_y)$. Thus, by analysing the partitioned norms we are able to determine how much the model is relying on certain sub-structures in the data, and as a result, how systematic the model is at all times during training.

Analytical expressions for these norms over training (which rely on both singular value dynamics and the structure of the singular vectors) are given in Appendix E Eqns. 17-20 due to space constraints.

---

[1] All experiments are run using the Jax library (Bradbury et al., 2018). Full code for reproducing all figures can be found at: `https://github.com/raillab/specialization_of_neural_modules`.

However, Figure 2(b),(d) depicts these dynamics for one specific dataset and Figure 3(a) summarizes the mappings between each input-output partition as well as which modes of variation contribute to each mapping. This leads to Observation 5.1 and the corresponding proof presented in Appendix H.1.

**Observation 5.1.** *For all datasets in the space it is impossible for a dense architecture to learn a systematic mapping between compositional components.*

*Proof Sketch:* For a network to be systematic the compositional output needs to be independent of the non-compositional input. Similarly, the compositional input should not influence the non-compositional output (to avoid the non-compositional output influencing the features learned by the systematic mapping). However, in Figure 3(a) we see that the mode $\pi_1$ contributes to all partitions of the network mapping. This shows that non-compositional components affect the compositional mappings ($\Gamma_x\Omega_y$-Norm$> 0$ and $\Omega_x\Gamma_y$-Norm$> 0$). Because of this the mapping is not systematic.

In sum, the implicit bias arising from depth, small random initialization, and gradient descent (Appendix I considers alternate learning rules to GD which do not change the outcome of the analysis) is insufficient to learn fully systematic mappings for any dataset in the space, since the mapping between compositional components shares a mode of variation with mappings involving non-compositional components. Thus, we note a necessary condition for systematicity: the mapping between lower-rank (compositional) sub-structure must be orthogonal (rely on different modes of variation) from the rest of the network mapping. In dense networks this is not the case as any correlation between sub-structures in the dataset will couple the mappings. Thus, we also consider whether modular architectures may form decoupled mappings between sub-structures.

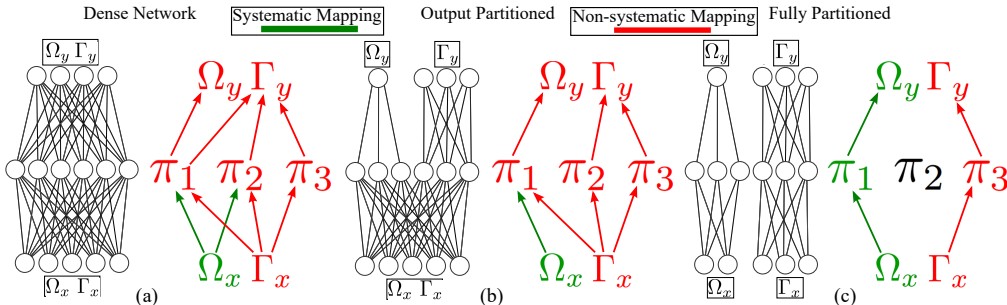

Figure 3: Graphical representation of the deep network mapping. The dynamical modes $\pi_1$, $\pi_2$, and $\pi_3$ contain contributions from compositional and non-compositional input components and they make contributions to compositional and non-compositional output components. Systematic portions of the mapping which rely only on compositional sub-structure are depicted as green. To be able to learn a systematic mapping all modes connected to the compositional output (input) component must not connect to the non-compositional input (output) component. (a) For the dense network we see that mode $\pi_1$ is the only mode which impacts the mapping to the compositional output (corresponds to the $\Omega_x\Omega_y$ and $\Gamma_x\Omega_y$ norms in Figure 2b being learned at the same time as the $\pi_1$ mode in Figure 2a), however $\pi_1$ also contributes to the non-compositional output mapping ($\Omega_x\Gamma_y$ and $\Gamma_x\Gamma_y$ norms also rise with $\pi_1$). Thus by learning the systematic mapping some non-systematic mapping is also being learned. (b and c) Impact of architectural biases. Architectures that partition compositional and non-compositional features in different ways with the corresponding graphical representation of the resulting network mappings. The output partitioned network is able to remove the impact of non-compositional output features on mode $\pi_1$ but does not result in a systematic mapping. Only the fully partitioned network achieves systematicity. Comparing to Figure 3(a) both graphical representations have less connections, particularly to/from the $\pi_1$ mode, reflecting the inductive bias imposed by modular architectures.

# 6 MODULARITY AND NETWORK ARCHITECTURE

We now turn to modularity and network architecture as a prominent approach for promoting systematicity in a network's mapping (Bahdanau et al., 2019b; Vani et al., 2021). Architectures such as NMNs (Andreas et al., 2016; Hu et al., 2017; 2018) learn re-configurable modules that implement specialized (lower-rank in accordance with Definition 3.1) aspects of a larger problem. By rearranging existing modules to process a novel input, they can generalize far beyond their training set. Here we investigate whether simple forms of additional architectural structure can aid in module

specialization and yield strong enough inductive biases to learn systematic mappings. From our results in Section 5 we know that a necessary inductive bias is for the partitioned norms to rely on different modes of variation in the data.

Instead of a dense network, we consider architectures in which compositional and non-compositional components are processed in separate processing streams, as depicted by the partitioned networks of Figure 3(b) and (c). The norm equations for the architecture which only partitions along output structure is summarized in Figure 3(b) (equations are presented in Appendix F). From this we make Observation 6.1 with a corresponding proof in Appendix H.2.

**Observation 6.1.** *Modularity does impose some form of bias towards systematicity but if any non-compositional structure is considered in the input then a systematic mapping will not be learned.*

*Proof Sketch:* We note that by separating the hidden layers from portions of the input and output space network modules will be learning on special cases of the datasets. For example, a module with no hidden layer connections to the non-compositional output components is learning on a dataset with $k_y = 0$. Thus, our norm equations from Section 5 still hold, just with the additional structure in the network limiting the potential datasets a module may see (the range of full input-output mappings is the same as before). In Figure 3(b) we see that the mode $\pi_1$ connects non-compositional input components to compositional output components ($\Gamma_x \Omega_y$-Norm$> 0$). Because of this the mapping is not systematic. We note that modularity has removed the connection between the compositional input and non-compositional output ($\Omega_x \Gamma_y$-Norm$= 0$). Because of this there is a step towards systematicity.

Finally, we consider the case where two modules are used and fully partition the compositional and non-compositional structure in the input-output mapping (the strategy employed by NMNs). The norms are summarized in Figure 3(c). From these equations we make Observation 6.2 and prove the result in Appendix H.3:

**Observation 6.2.** *The fully partitioned network achieves systematicity by learning the lower-rank (compositional) sub-structure in isolation of the rest of the mapping.*

*Proof Sketch:* In Figure 3(c) we see that compositional input only connects to compositional output ($\Gamma_x \Omega_y$-Norm$= 0$). Similarly, non-compositional input only connects to non-compositional output ($\Omega_x \Gamma_y$-Norm$= 0$). The portion of the network mapping from compositional input to compositional output is solving a lower rank problem than if the full feature spaces were considered. Because of this the network will generalise on this portion of the mapping and is systematic in terms of Definition 3.1.

Finally, partitioning hidden layers is not possible for shallow networks, and so we are unable to obtain partial benefits from modularity with shallow networks. Thus, shallow networks also require the perfect architectural biases to partition the input and output structure to become systematic. Hence we find that architectural biases (modularity) can enforce systematicity, when the bias perfectly segregates compositional and non-compositional information. How to achieve the correct segregation remains an open question in the NMNs literature, where reinforcement learning is often used (Hu et al., 2017; 2018), and we do not discuss this here. We merely aim to show the necessity of modularity, which has been hypothesized (Hadley, 1994) and demonstrated (Andreas et al., 2016) but not theoretically shown. In Appendix G we consider the effects of imperfect output partitions and find that if any non-compositional components are grouped with compositional components then the network will behave the same as in Section 5 and no longer be able to learn a systematic mapping. Taken together, the observations of this section demonstrate the difficulty of learning specialized neural modules, as if any information not of the desired lower-rank sub-structure is seen by a module then it will fail to specialize. We now empirically verify if this generalises to a more naturalistic setting and non-linear architectures.

# 7 COMPOSITIONAL MNIST (CMNIST)

To evaluate how well our results generalize to non-linear networks and more complex datasets, in this section we train a deep Convolutional Neural Network (CNN) to learn a compositional variant of MNIST (CMNIST) shown in Figure 4a. In this dataset three digits from MNIST are stacked horizontally, resulting in a value between 0 and 999. The systematic output encodes each digit in a 10-way one-hot vector, resulting in a 30-dimensional vector. The non-systematic output encodes the number as a whole with a 1000-dimensional one-hot vector. This task is similar to the SVHN dataset (Netzer et al., 2011) with added non-systematic labels.

Inputs     Network     Labels

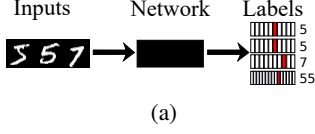

(a)

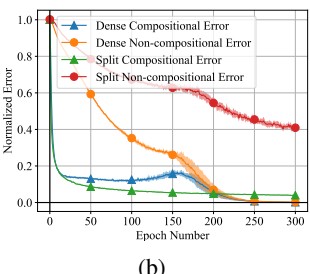

(b)

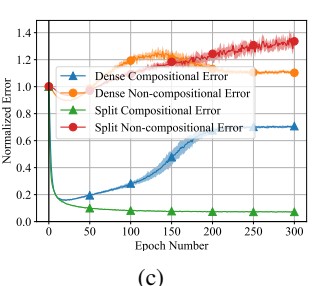

(c)

Figure 4: (a) Visual description of the CMNIST task, (b-c) normalized training loss (b) and test loss (c) of a deep CNN with ReLU activation on the Compositional-MNIST dataset averaged over 10 runs. Error bars reflect one standard-deviation.

We compare results of using a single dense CNN and a split CNN, in which two parallel sets of convolutional layers with half the convolutional filters of the dense network each connect separately to systematic and non-systematic labels. Full details of the two network architectures and hyper-parameters used for the CMNIST experiments are given in Appendix J. The effect predicted by our theoretical work is that the non-compositional 1000-dimensional one-hot label will interfere with the network's learned hidden representations for the compositional 30-dimensional label. This will decrease the compositional generalizability of the network on this structured portion of the output. This prediction is shown to be true in Figure 4, which shows the mean-squared error for the compositional and non-compositional network predictions over the course of training, normalized so that the initial error is at $1.0$. Firstly, in dense networks the error of the compositional mapping is tied to the error of the non-compositional mapping. This is seen in Figure 4b where the blue curve cannot converge until the orange curve has become sufficiently low. This effect is not observed with the split network. Secondly, comparing Figure 4b and 4c we demonstrate the lack of generalization which occurs when using non-compositional components. This is seen as the orange and red curves achieving a lower training error while the test error increases. Lastly, again by comparing Figure 4b and 4c, we see that the compositional mapping of the split architecture is the only mapping which generalizes well (near 0 training and test error). Thus, even in this more complex setting, we see the negative effect a shared hidden layer has on the generalization of the network (comparing the blue and green curves).

## 8 DISCUSSION

In this work we have theoretically and empirically studied the ability of simple NN modules to specialize and acquire systematic knowledge. We found that this ability is challenging even in our simple setting. Neither implicit biases in learning dynamics, nor all but the most stringent task-specific modularity, caused networks to exploit compositional sub-structure in the data. Our results complement recent empirical studies, helping to highlight the complex factors influencing systematic generalization. For example, in more complex datasets such as CLEVR (Johnson et al., 2017), a module which should specialise to identifying the colour red would not specialise if a Ferrari logo (a non-compositional identifying feature) was present in many of these images – the network would not identify sub-structure in $\Sigma^x$. Similarly, if many images containing red features mapped to a particular label, for example the "car" label, this would also affect which features the module specialised to. When the module is then used to "find the red ball" it would be looking for stereotypical car features to identify the colour red – it does not identify sub-structure in $\Sigma^{yx}$. Thus, in more natural settings our proposed notion of rank extends to describing the complexity of the correlations being learned. Identifying and using the colour red in images is less complex than identifying and using the presence of a Ferrari; requiring many different input features (higher rank $\Sigma^x$) to be identified and correlated to a certain label (higher rank $\Sigma^{yx}$). Importantly for Definition 3.1, if a task is specific to Ferraris, then having a specialised Ferrari module is useful (all test images will also contain a Ferrari). But in general scene description tasks it is not. Consequently, Definition 3.1 describes systematicity in terms of the task structure at hand: $\Sigma^x$ and $\Sigma^{yx}$. Thus, as our theoretical module-centric perspective displays, modules must be allocated perfectly; such that the only consistent correlation presented to a module is the one it must specialise to. A natural question emerges: which inductive biases are flexible enough to achieve systematicity without being tailored to a specific problem? A neuroscientific perspective is that sparsity plays a role in producing systematic representations, particularly in the cerebellum (Cayco-Gajic & Silver, 2019). Thus, exploring the utility of sparsity for systematicity is an important direction of future work. We hope that the formal perspective provided in this work will lead to greater understanding of these phenomena and aid the design of improved, modular, learning systems.

ACKNOWLEDGMENTS

We would like to thank the reviewers for their time and careful consideration of this manuscript. We sincerely appreciate all their valuable comments and suggestions, which helped us in improving the quality of the manuscript. This work was supported by a Sir Henry Dale Fellowship from the Wellcome Trust and Royal Society (216386/Z/19/Z) to A.S., and the Sainsbury Wellcome Centre Core Grant from Wellcome (219627/Z/19/Z) and the Gatsby Charitable Foundation (GAT3755). D.J. is a Google PhD Fellow and Commonwealth Scholar. A.S. and B.R. are CIFAR Azrieli Global Scholars in the Learning in Machines & Brains program.

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

## A    MOTIVATING EXAMPLE

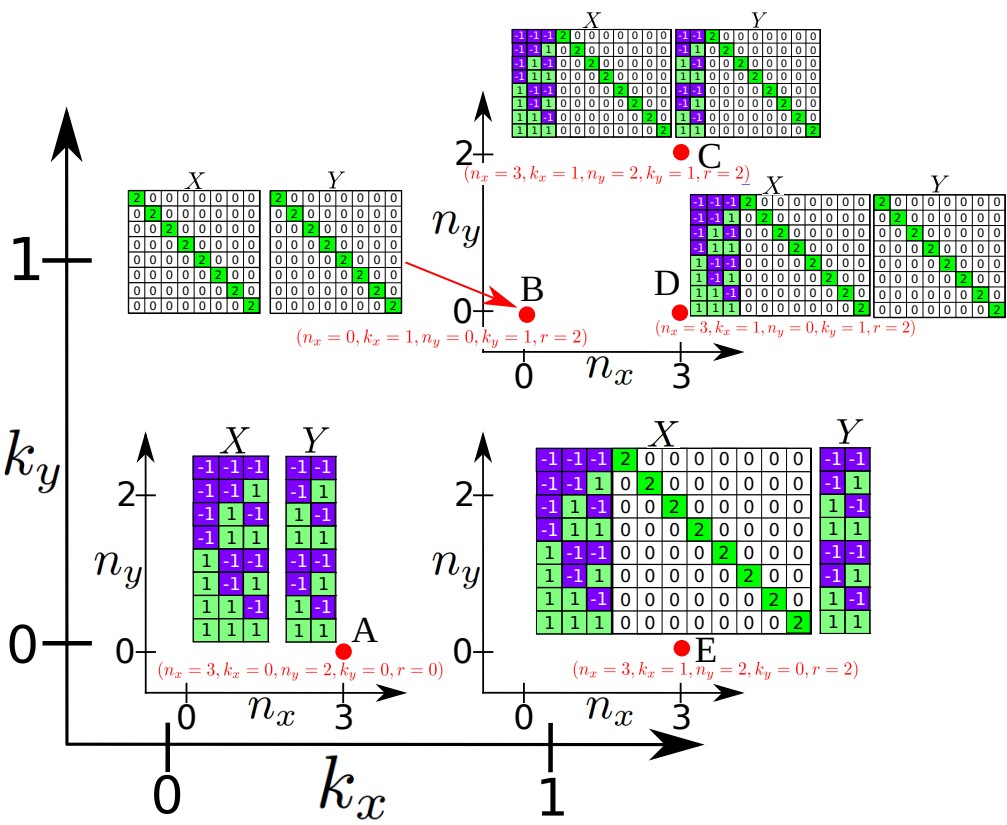

Figure 5: Example of the space of datasets with five exemplar datasets. Datasets range in the number of compositional components (the inner axes of $n_x$ and $n_y$) and the number of non-compositional identity blocks (outer axis of $k_x$ and $k_y$). Note $n_x$ and $n_y$ vary the number of compositional features, while $k_x$ and $k_y$ vary the number of non-compositional blocks of features. In this case the scale parameter of the identity blocks $r$ is fixed at 2.

In this section we present a motivating example which is designed to provide more intuition on the space of datasets presented in Section 3 and our definition of systematicity: Definition 3.1. We consider five different exemplar datasets from the space, depicted in Figure 5, present the empirical training and test accuracy of a simulated linear network and take note of the generalisation of each trained network. Ultimately, we demonstrate that systematic generalisation, in accordance with Definition 3.1, does not just imply an ability to correctly label unseen data points; but also relies on identifying sub-structure in the feature space to do so.

For these simulated trainings we use the first 3 out of a total 8 data points as training data, leaving the remaining 5 data points as a test dataset. We term these as the "training dataset" and "test dataset" respectively and term the combination of both with all 8 data points as the "full dataset". All networks contain 50 hidden neurons, are trained with a learning rate of 0.02 and are initialised with a 0 centered normal distribution with standard deviation 0.001. We emphasise that we are presenting just five possible datasets in our entire space and have chosen these datasets to make the images and arguments convenient and intuitive. Whether a trained network will effectively generalise to the test dataset

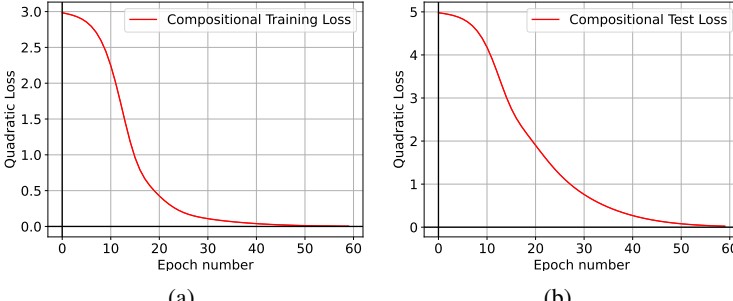

(a)                                                             (b)

Figure 6: Training (a) and Test (b) loss of a linear network trained on dataset A ($n_x = 3, k_x = 0, n_y = 2, k_y = 0, r = 0$) with a $3 : 5$ train-test split of the dataset averaged over $50$ runs.

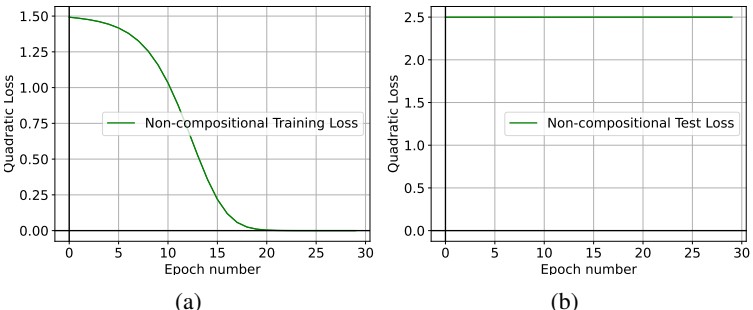

(a)                                                             (b)

Figure 7: Training (a) and Test (b) loss of a linear network trained on dataset B ($n_x = 0, k_x = 1, n_y = 0, k_y = 1, r = 2$) with a $3 : 5$ train-test split of the dataset averaged over $50$ runs.

depends on the rank of the full dataset and number of training data points it observes. Simply put: the network must be trained on as many (unique) data points as the rank of the full dataset. We now demonstrate this point with the examples and simulations.

Take for example our first dataset with only compositional input and output components corresponding to point A in Figure 5 ($n_x = 3, k_x = 0, n_y = 2, k_y = 0, r = 0$). In this case there are $8$ data points in the full dataset but the covariance matrices $\Sigma^x \in \mathbb{R}^{3\times3}$, $\Sigma^{yx} \in \mathbb{R}^{2\times3}$ have ranks of 3 and 2 respectively. Thus, by using a training set of 3 data points the network will effectively learn the input-output mapping: $W^2 W^1 = \Sigma^{yx}\Sigma^{x-1}$. As a result the network has learned the correct input-output mapping for the full dataset. Thus, the network will generalise to the test dataset. This is displayed for a simulated training in Figure 6. Importantly, even though the network generalises, this is not yet systematic generalisation as we have defined it in Definition 3.1.

In comparison, consider a dataset with $8$ data points of only non-compositional input and output components corresponding to point B in Figure 5 ($n_x = 0, k_x = 1, n_y = 0, k_y = 1, r = 2$). In this case the covariance matrices $\Sigma^x \in \mathbb{R}^{8\times8}$, $\Sigma^{yx} \in \mathbb{R}^{8\times8}$ both have a rank of 8 and if a training dataset of 3 data points is use then network will fail to learn the covariance matrix (and mapping) appropriate for the full dataset. As a result the network trained on the training set will not generalise. Since there are only 8 data points to learn the rank 8 covariance matrices, generalisation is not possible for this dataset (the network needs to be trained on the full dataset). Figure 7 displays the simulation for this case. Note that the test error does not decrease during training.

Now consider putting the two settings together ($n_x = 3, k_x = 1, n_y = 2, k_y = 1, r = 2$) which corresponds to point C in Figure 5. We have the compositional dataset being appended with a non-compositional dataset to arrive at a dataset of $8$ data points with both compositional and non-compositional sub-structure (on both the input and the output). If we were to calculate the covariance matrices ($\Sigma^x \in \mathbb{R}^{11\times11}$, $\Sigma^{yx} \in \mathbb{R}^{10\times11}$) for this full dataset we would see that both have a rank of 8 due to the non-compositional sub-structure being of rank 8. Thus, for the same reasons as above generalisation from a training dataset of 3 data points is not possible. This case is displayed in Figure 8. Note than, even though compositional features are present on the input and output the network does not generalise well, even when considering the test performance for compositional

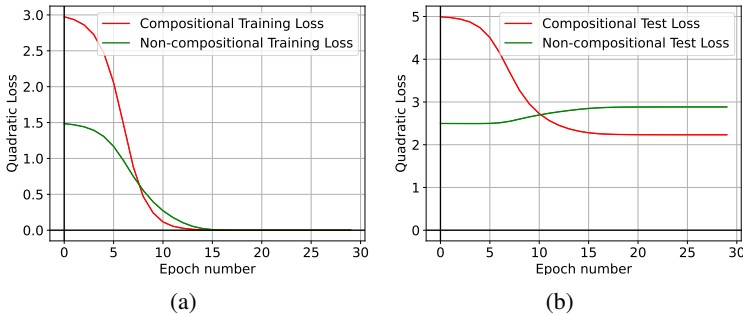

Figure 8: Training (a) and Test (b) loss of a linear network trained on dataset C ($n_x = 3, k_x = 1, n_y = 2, k_y = 1, r = 2$) with a $3 : 5$ train-test split of the dataset averaged over 50 runs.

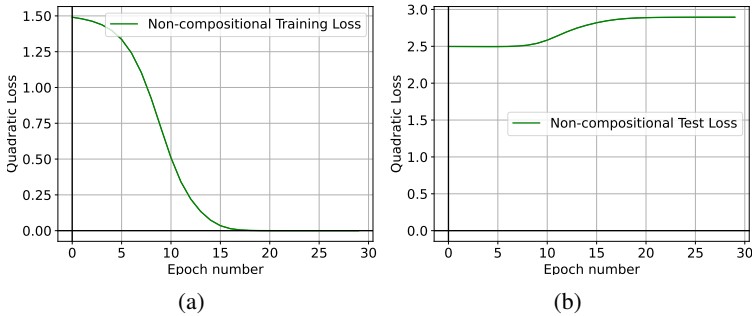

Figure 9: Training (a) and Test (b) loss of a linear network trained on dataset D ($n_x = 3, k_x = 1, n_y = 0, k_y = 1, r = 2$) with a $3 : 5$ train-test split of the dataset averaged over 50 runs.

outputs in isolation. However, we know that if we were to consider the compositional sub-structure independently it would be a rank 3 problem and would support generalisation. The easier task of mapping compositional input to compositional output (the first dataset we considered) exists within the larger problem of learning the full input-output mapping of this combined dataset. This was just obscured by the inclusion of the non-compositional components. **This** is our notion of systematic generalisation: identifying sub-structure in a larger problem (which does not support generalisation) such that we can learn the sub-structure separately to support generalisation. Thus, our definition of systematicity requires finding the sub-structure as much as exploiting it for generalisation.

We display two more cases corresponding to points D ($n_x = 3, k_x = 1, n_y = 0, k_y = 1, r = 2$) and E ($n_x = 3, k_x = 1, n_y = 2, k_y = 0, r = 2$) in Figure 5. The simulations for each dataset are presented in Figures 9 and 10 respectively. In both cases non-compositional input components are present and the network does not learn to rely solely on the lower-rank compositional sub-structure. As a result the network does not systematically generalise.

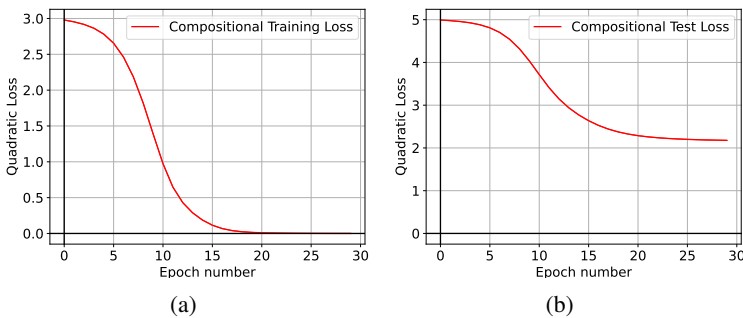

Figure 10: Training (a) and Test (b) loss of a linear network trained on dataset E ($n_x = 3, k_x = 1, n_y = 2, k_y = 0, r = 2$) with a $3 : 5$ train-test split of the dataset averaged over 50 runs.

## B  RANK OF COMPOSITIONAL DATASET SUB-STRUCTURES

In this section we provide the calculations of the chance of sampling a full-rank matrix for the compositional partition of the input and output space. We will begin by considering the case where the number of compositional input and output features are both $d = 3$. We emphasize that this section aims to demonstrate the benefit of relying on compositional sub-structure to learn a network mapping and is not meant to be general to all forms of binary matrices. Since we are using three input and output features there are $2^3 = 8$ unique data points in the input and output. We are concerned then with the rank of the empirical covariance matrix $\hat{\Sigma}^{yx} \in \mathbb{R}^{3\times 3}$. We use the empirical covariance matrix since we only use samples of the dataset to train a network and not the full dataset.

For a neural network to learn a generalizable mapping it needs to learn from a batch of data with a full-rank covariance matrix that has the same rank as the full dataset. If this is the case then the network will learn the same mapping from the training sample as it would on the full dataset and will generalize to all unseen data points from the same space. We note that for sample sizes of $n = 1$ and $n = 2$ it is impossible to obtain a full-rank empirical covariance. Thus, we first consider a sample size of $n = 3$. In the 3-dimensional case the only way to obtain a singular matrix from 3 samples is if the opposite data point to an earlier data point is sampled. For example, for the data point $[-1, -1, 1]$ the opposite will be $[1, 1, -1]$. With this observation we can determine the probability of not sampling an opposite data point as follows:

$$P(fullrank|n=3, d=3) = P(throw_3|throw_1, throw_2)P(throw_2|throw_1)P(throw_1)$$
$$= (4/6)(6/7)(8/8) = 0.57$$

For a sample of $n = 4$ data points a similar procedure applies, except we need to account for the fact that we are able to sample one opposite data point now. Thus, there are four cases: the second data point is an opposite, the third data point is an opposite, the fourth data point is an opposite, or there are no opposites (we also drop $P(throw_1) = 8/8$ to lighten notation):

$$P(fullrank|n=4, d=3) =$$
$$P(throw_4|throw_1, throw_2, throw_3)P(throw_3|throw_1, throw_2)P(\neg throw_2|throw_1)$$
$$+ P(throw_4|throw_1, throw_2, throw_3)P(\neg throw_3|throw_1, throw_2)P(throw_2|throw_1)$$
$$+ P(\neg throw_4|throw_1, throw_2, throw_3)P(throw_3|throw_1, throw_2)P(throw_2|throw_1)$$
$$+ P(throw_4|throw_1, throw_2, throw_3)P(throw_3|throw_1, throw_2)P(throw_2|throw_1)$$
$$= (4/5)(6/6)(1/7) + (4/5)(2/6)(6/7) + (3/5)(4/6)(6/7) + (2/5)(4/6)(6/7)$$
$$= 0.91$$

For a sample size of $n = 5$ or more we are guaranteed to have at least 3 linearly independent data points and a full-rank covariance matrix. Thus, the network module trained on this sub-structure will always generalize. A similar set of calculation can be done for the case of 4 compositional features and the same conclusions hold. The calculations, however, become large due to larger sample sizes being used and more possibilities to sample linearly dependent vectors. Thus, we present the probabilities from empirical results for the 3 features and 4 feature cases in Tables 1 and 2 respectively. For these empirical results 5000 samplings from the dataset are used for each sample size to calculate a covariance matrix and determine if it is singular.

| Sample Size | Probability of Full-Rank Covariance |
|:---:|:---:|
| 3 | 57.5% |
| 4 | 91.9% |
| {5,...,8} | 100% |

Table 1: Probability of sampling a full-rank covariance matrix for the binary compositional component of the input and output with 3 features each ($n_x = n_y = 3$)

| Sample Size | Probability of Full-Rank Covariance |
|:---:|:---:|
| 4 | 29.74% |
| 5 | 84.76% |
| 6 | 93.82% |
| 7 | 99.2% |
| 8 | 99.72% |
| {9,...,16} | 100% |

Table 2: Probability of sampling a full-rank covariance matrix for the binary compositional component of the input and output with 4 features each ($n_x = n_y = 4$)

## C  LEARNING DYNAMICS IN DEEP LINEAR NETWORKS

The dynamics of learning for shallow and deep linear networks are derived in Saxe et al. (2019). We state full details of our specific setting here. We train a linear network with one hidden layer to minimize the quadratic loss $L(W^1, W^2) = \frac{1}{2^{n_x}}||Y - W^2W^1X||_2^2$ using gradient descent. This gives the learning rules $E[\Delta W^1] = \frac{\epsilon}{2^{n_x}}W^{2^T}(Y - W^2W^1X)X^T$ and $E[\Delta W^2] = \frac{\epsilon}{2^{n_x}}(Y - W^2W^1X)(W^1X)^T$. By using a small learning rate $\epsilon$ and taking the continuous time limit, the mean change in weights is given by $\tau\frac{d}{dt}W^1 = W^{2^T}(\Sigma^{yx} - W^2W^1\Sigma^x)$ and $\tau\frac{d}{dt}W^2 = (\Sigma^{yx} - W^2W^1\Sigma^x)W^{1^T}$ where $\Sigma^x = E[XX^T]$ is the input correlation matrix, $\Sigma^{yx} = E[YX^T]$ is the input-output correlation matrix and $\tau = \frac{1}{2^{n_x}\epsilon}$. Here, $t$ measures units of learning epochs. It is helpful to note that since we are using a small learning rate the full batch gradient descent and stochastic gradient descent dynamics will be the same. Saxe et al. (2019) has shown that the learning dynamics depend on the singular value decomposition of $\Sigma^{yx} = USV^T = \sum_{\alpha=1}^{2^n}\lambda_\alpha u^\alpha v^{\alpha^T}$ and $\Sigma^x = VDV^T = \sum_{\alpha=1}^{2^{n_x}}\delta_\alpha u^\alpha v^{\alpha^T}$. To solve for the dynamics we require that the right singular vectors $V$ of $\Sigma^{yx}$ are also the singular vectors of $\Sigma^x$. This is the case for any dataset in our space, as shown in Appendix D. Note, we assume that the network has at least $2^{n_x}$ hidden neurons (the number of singular values in the input-output covariance matrix) so that it can learn the desired mapping perfectly. If this is not the case then the model will learn the top $n_h$ singular values of the input-output mapping where $n_h$ is the number of hidden neurons (Saxe et al., 2014). Given the SVDs of the two correlation matrices the learning dynamics can be described explicitly as $W^2(t)W^1(t) = UA(t)V^T = \sum_{\alpha=1}^{2^{n_x}}\pi_\alpha(t)u^\alpha v^{\alpha T}$ where $A(t)$ is the effective singular value matrix of the network's mapping. The trajectory of each singular value in $A(t)$ is described as $\pi_\alpha(t) = \frac{\lambda_\alpha/\delta_\alpha}{1-(1-\frac{\lambda_\alpha}{\delta_\alpha\pi_0})\exp(\frac{-2\lambda_\alpha}{\tau}t)}$. From these dynamics it is helpful to note that the time-course of the trajectory is only dependent on the $\Sigma^{yx}$ singular values. Thus, $\Sigma^x$ affects the stable point of the network singular values but not the time-course of learning.

We have chosen our datasets to be simple and interpretable for clarity and tractability. However, fundamentally, our analysis with deep linear networks applies to a more general situation for which other theoretical techniques relying on random initialisation fail (Geiger et al., 2020; Jacot et al., 2018; Sirignano & Spiliopoulos, 2020). In particular, consider the broader space of datasets in which compositional inputs, compositional outputs, non-compositional inputs, and non-compositional outputs are each rotated by individual orthogonal mappings. As one specific example, consider the dataset formed by applying a permutation to the compositional outputs. This change affects only the singular vectors of the task (also permuting them relative to the original task), not the singular values. This dataset can no longer be solved by learning an identity transformation (which is a possible solution for all datasets in the original space), as the network must learn to implement the required permutation–but it will do so with identical learning dynamics and systematicity properties. A random initialisation cannot know this desired permutation, which can only be learned through error feedback. This, further, motivates the use of the linear network dynamics which incorporates feature learning into the analysis.

## D SINGULAR VALUE DECOMPOSITION EQUATIONS

In this section we provide the general formulas for the singular value decomposition of the $\Sigma^x$ and $\Sigma^{yx}$ covariance matrices for any dataset in the space of datasets. As stated in Section 4 the right singular vectors of $\Sigma^{yx}$ must match the singular vectors of $\Sigma^x$ which is the $V$ matrix below. Thus, $\Sigma^x = VDV^T$ and $\Sigma^{yx} = USV^T$.

Let:
$A = (\Omega_y \Omega_x^T)^T \Omega_y \Omega_x^T$
$B = \Omega_y^T \Omega_y \Omega_x^T$
$C = \Omega_x^T \Omega_x \Omega_x^T$
$THP^T = (\frac{1}{k_x k_y})^{\frac{1}{2}} \mathbf{I}_{2^{n_x} \times 2^{n_x}} - (\frac{1}{k_x k_y})^{\frac{1}{2}}(\frac{1}{2^{n_x}})\Omega_x^T \Omega_x$
Where $THP^T$ is the SVD of $(\frac{1}{k_x k_y})^{\frac{1}{2}} \mathbf{I}_{2^{n_x} \times 2^{n_x}} - (\frac{1}{k_x k_y})^{\frac{1}{2}}(\frac{1}{2^{n_x}})\Omega_x^T \Omega_x$, and $H = (\frac{1}{k_x k_y})^{\frac{1}{2}} \mathbf{I}_{2^{n_x} \times 2^{n_x}}$.
Then the following are the matrix formulas for the components of the SVD for $\Sigma^{yx}$ and $\Sigma^x$.

$$U = \begin{bmatrix} \left(\frac{1}{2^{n_x}(k_y r^2 + 2^{n_x})}\right)^{\frac{1}{2}} \Omega_y \Omega_x^T & \mathbf{0}_{\mathbf{n_y} \times \mathbf{k_x} 2^{\mathbf{n_x}}} \\ \left(\frac{r^2}{2^{3n_x}(k_y r^2 + 2^{n_x})}\right)^{\frac{1}{2}} B + \left(\frac{r^2}{2^{3n_x}(k_y r^2)}\right)^{\frac{1}{2}}(C - B) & (\frac{1}{k_y})^{\frac{1}{2}} T \end{bmatrix} \tag{13}$$

$$V^T = \begin{bmatrix} \left(\frac{2^{n_x}}{(k_x r^2 + 2^{n_x})}\right)^{\frac{1}{2}} \mathbf{I}_{\mathbf{n_x} \times \mathbf{n_x}} & \left(\frac{r^2}{2^{n_x}(k_x r^2 + 2^{n_x})}\right)^{\frac{1}{2}} \Omega_x \\ \mathbf{0}_{\mathbf{k_x} 2^{\mathbf{n_x}} \times \mathbf{n_x}} & (\frac{1}{k_x})^{\frac{1}{2}} P^T \end{bmatrix} \tag{14}$$

$$S = \begin{bmatrix} \left(\frac{(k_x r^2 + 2^{n_x})(k_y r^2 + 2^{n_x})}{2^{6n_x}}\right)^{\frac{1}{2}} A + \left(\frac{(k_x r^2 + 2^{n_x})(k_y r^2)}{2^{2n_x}}\right)^{\frac{1}{2}} (I - \frac{1}{2^{2n_x}} A) & \mathbf{0}_{\mathbf{n_x} \times \mathbf{k_x} 2^{\mathbf{n_x}}} \\ \mathbf{0}_{\mathbf{k_x} 2^{\mathbf{n_x}} \times \mathbf{n_x}} & (k_x k_y)^{\frac{1}{2}} \frac{r^2}{2^{n_x}} \mathbf{I}_{\mathbf{k_x} 2^{\mathbf{n_x}} \times \mathbf{k_x} 2^{\mathbf{n_x}}} \end{bmatrix} \tag{15}$$

$$D = \begin{bmatrix} \left(\frac{(k_x r^2 + 2^{n_x})}{2^{n_x}}\right)^{\frac{1}{2}} \mathbf{I}_{\mathbf{n_x} \times \mathbf{n_x}} & \mathbf{0}_{\mathbf{n_x} \times \mathbf{k_x} 2^{\mathbf{n_x}}} \\ \mathbf{0}_{\mathbf{k_x} 2^{\mathbf{n_x}} \times \mathbf{n_x}} & \frac{k_x r^2}{2^{n_x}} \mathbf{I}_{2^{\mathbf{n_x}} \times 2^{\mathbf{n_x}}} \end{bmatrix} \tag{16}$$

We note that each distinct singular value occurs multiple times in the dataset: $\lambda_1$ has multiplicity $n_x - n_y$, $\lambda_2$ has multiplicity $n_y$, and $\lambda_3$ has multiplicity $2^{n_x} - n_x$. Correspondingly $\delta_1$ has multiplicity $n_x$ and $\delta_2$ has multiplicity $2^{n_x} - n_x$.

### D.1 PROVING THE CORRECTNESS OF THE SVD

Proving the correctness of the above SVD is easier than deriving it. Thus we do so here by showing that $\Sigma^{yx} = USV^T$:

$$SV^T = \begin{bmatrix} \left(\frac{(k_x r^2 + 2^{n_x})(k_y r^2 + 2^{n_x})}{2^{6n_x}}\right)^{\frac{1}{2}} A + \left(\frac{(k_x r^2 + 2^{n_x})(k_y r^2)}{2^{2n_x}}\right)^{\frac{1}{2}} (I - \frac{1}{2^{2n_x}} A) & \mathbf{0}_{\mathbf{n_x} \times \mathbf{k_x} 2^{\mathbf{n_x}}} \\ \mathbf{0}_{\mathbf{k_x} 2^{\mathbf{n_x}} \times \mathbf{n_x}} & (k_x k_y)^{\frac{1}{2}} \frac{r^2}{2^{n_x}} \mathbf{I}_{\mathbf{k_x} 2^{\mathbf{n_x}} \times \mathbf{k_x} 2^{\mathbf{n_x}}} \end{bmatrix}$$

$$\begin{bmatrix} \left(\frac{2^{n_x}}{(k_x r^2 + 2^{n_x})}\right)^{\frac{1}{2}} \mathbf{I}_{\mathbf{n_x} \times \mathbf{n_x}} & \left(\frac{r^2}{2^{n_x}(k_x r^2 + 2^{n_x})}\right)^{\frac{1}{2}} \Omega_x \\ \mathbf{0}_{\mathbf{k_x} 2^{\mathbf{n_x}} \times \mathbf{n_x}} & (\frac{1}{k_x})^{\frac{1}{2}} P^T \end{bmatrix}$$

$$SV^T = \begin{bmatrix} \left(\frac{k_y r^2 + 2^{n_x}}{2^{5n_x}}\right)^{\frac{1}{2}} A + \left(\frac{k_y r^2}{2^{n_x}}\right)^{\frac{1}{2}} (I - \frac{1}{2^{2n_x}} A) & \left(\frac{r^2(k_y r^2 + 2^{n_x})}{2^{7n_x}}\right)^{\frac{1}{2}} A\Omega_x + \left(\frac{k_y r^4}{2^{3n_x}}\right)^{\frac{1}{2}} (I - \frac{1}{2^{2n_x}} A)\Omega_x \\ \mathbf{0}_{\mathbf{n_x} \times \mathbf{k_x} 2^{\mathbf{n_x}}} & (k_y)^{\frac{1}{2}} \frac{r^2}{2^{n_x}} P^T \end{bmatrix}$$

$$USV^T = \begin{bmatrix} \left(\frac{1}{2^{n_x}(k_y r^2 + 2^{n_x})}\right)^{\frac{1}{2}} \Omega_y \Omega_x^T & \mathbf{0}_{\mathbf{n_y} \times \mathbf{k_x} 2^{\mathbf{n_x}}} \\ \left(\frac{r^2}{2^{3n_x}(k_y r^2 + 2^{n_x})}\right)^{\frac{1}{2}} B + \left(\frac{r^2}{2^{3n_x}(k_y r^2)}\right)^{\frac{1}{2}} (C - B) & (\frac{1}{k_y})^{\frac{1}{2}} T \end{bmatrix}$$

$$\begin{bmatrix} \left(\frac{k_y r^2 + 2^{n_x}}{2^{5n_x}}\right)^{\frac{1}{2}} A + \left(\frac{k_y r^2}{2^{n_x}}\right)^{\frac{1}{2}} (I - \frac{1}{2^{2n_x}} A) & \left(\frac{r^2(k_y r^2 + 2^{n_x})}{2^{7n_x}}\right)^{\frac{1}{2}} A\Omega_x + \left(\frac{k_y r^4}{2^{3n_x}}\right)^{\frac{1}{2}} (I - \frac{1}{2^{2n_x}} A)\Omega_x \\ \mathbf{0}_{\mathbf{n_x} \times \mathbf{k_x} 2^{\mathbf{n_x}}} & (k_y)^{\frac{1}{2}} \frac{r^2}{2^{n_x}} P^T \end{bmatrix}$$

From here we will solve each quadrant separately ($Q_1$ to $Q_4$ below). Firstly, we mention some useful identities:

$$A = (\Omega_y \Omega_x^T)^T \Omega_y \Omega_x^T$$

$$B = \Omega_y^T \Omega_y \Omega_x^T$$

$$C = \Omega_x^T \Omega_x \Omega_x^T$$

$$\Omega_y \Omega_x^T (\Omega_y \Omega_x^T)^T = 2^{2n_x} I_{n_y \times n_y}$$

$$\Omega_y \Omega_x^T A = \Omega_y \Omega_x^T (\Omega_y \Omega_x^T)^T \Omega_y \Omega_x^T = 2^{2n_x} \Omega_y \Omega_x^T$$

$$BA = \Omega_y^T \Omega_y \Omega_x^T (\Omega_y \Omega_x^T)^T \Omega_y \Omega_x^T = \Omega_y^T 2^{2n_x} \Omega_y \Omega_x^T = 2^{2n_x} B$$

$$CA = \Omega_x^T \Omega_x \Omega_x^T (\Omega_y \Omega_x^T)^T \Omega_y \Omega_x^T = \Omega_x^T 2^{2n_x} \Omega_y \Omega_x^T = 2^{2n_x} B$$

We now solve for each quadrant of the $USV^T$ matrix:

$$
\begin{aligned}
Q_1 &= \left( \frac{1}{2^{6n_x}} \right)^{\frac{1}{2}} \Omega_y \Omega_x^T A + \left( \frac{k_y r^2}{2^{2n_x}(k_y r^2 + 2^{n_x})} \right)^{\frac{1}{2}} \Omega_y \Omega_x^T (I - \frac{1}{2^{2n_x}} A) \\
&= \frac{1}{2^{n_x}} \Omega_y \Omega_x^T + \left( \frac{k_y r^2}{2^{2n_x}(k_y r^2 + 2^{n_x})} \right)^{\frac{1}{2}} \Omega_y \Omega_x^T - \left( \frac{k_y r^2}{k_y r^2 + 2^{n_x}} \right)^{\frac{1}{2}} \Omega_y \Omega_x^T \\
&= \frac{1}{2^{n_x}} \Omega_y \Omega_x^T
\end{aligned}
$$

$$
\begin{aligned}
Q_2 &= \left( \frac{r^2}{2^{8n_x}} \right)^{\frac{1}{2}} \Omega_y \Omega_x^T A \Omega_x + \left( \frac{k_y r^4}{2^{4n_x}(k_y r^2 + 2^{n_x})} \right)^{\frac{1}{2}} \Omega_y \Omega_x^T (I - \frac{1}{2^{2n_x}} A) \Omega_x \\
&= \frac{r}{2^{4n_x}} \Omega_y \Omega_x^T A \Omega_x + \left( \frac{k_y r^4}{2^{4n_x}(k_y r^2 + 2^{n_x})} \right)^{\frac{1}{2}} \Omega_y \Omega_x^T \Omega_x - \left( \frac{k_y r^4}{2^{8n_x}(k_y r^2 + 2^{n_x})} \right)^{\frac{1}{2}} \Omega_y \Omega_x^T A \Omega_x \\
&= \frac{r}{2^{2n_x}} \Omega_y \Omega_x^T \Omega_x + \left( \frac{k_y r^4}{2^{4n_x}(k_y r^2 + 2^{n_x})} \right)^{\frac{1}{2}} \Omega_y \Omega_x^T \Omega_x - \left( \frac{k_y r^4}{2^{4n_x}(k_y r^2 + 2^{n_x})} \right)^{\frac{1}{2}} \Omega_y \Omega_x^T \Omega_x \\
&= \frac{r}{2^{2n_x}} \Omega_y \Omega_x^T \Omega_x = \frac{r}{2^{n_x}} \Omega_y
\end{aligned}
$$

$$
\begin{aligned}
Q_3 &= \left[ \left( \frac{r^2}{2^{3n_x}(k_y r^2 + 2^{n_x})} \right)^{\frac{1}{2}} B + \left( \frac{r^2}{2^{3n_x}(k_y r^2)} \right)^{\frac{1}{2}} (C - B) \right] \left[ \left( \frac{k_y r^2 + 2^{n_x}}{2^{5n_x}} \right)^{\frac{1}{2}} A + \left( \frac{k_y r^2}{2^{n_x}} \right)^{\frac{1}{2}} (I - \frac{1}{2^{2n_x}} A) \right] \\
&= \left[ \left( \frac{r^2}{2^{3n_x}(k_y r^2 + 2^{n_x})} \right)^{\frac{1}{2}} B + \left( \frac{r^2}{2^{3n_x}(k_y r^2)} \right)^{\frac{1}{2}} (C - B) \right] \left( \frac{k_y r^2 + 2^{n_x}}{2^{5n_x}} \right)^{\frac{1}{2}} A \\
&\quad + \left[ \left( \frac{r^2}{2^{3n_x}(k_y r^2 + 2^{n_x})} \right)^{\frac{1}{2}} B + \left( \frac{r^2}{2^{3n_x}(k_y r^2)} \right)^{\frac{1}{2}} (C - B) \right] \left( \frac{k_y r^2}{2^{n_x}} \right)^{\frac{1}{2}} (I - \frac{1}{2^{2n_x}} A) \\
&= \left( \frac{r^2}{2^{8n_x}} \right)^{\frac{1}{2}} BA + \left( \frac{r^2(k_y r^2 + 2^{n_x})}{2^{8n_x}(k_y r^2)} \right)^{\frac{1}{2}} (C - B) A \\
&\quad + \left( \frac{k_y r^4}{2^{4n_x}(k_y r^2 + 2^{n_x})} \right)^{\frac{1}{2}} B (I - \frac{1}{2^{2n_x}} A) + \left( \frac{r^2}{2^{4n_x}} \right)^{\frac{1}{2}} (C - B)(I - \frac{1}{2^{2n_x}} A) \\
&= \left( \frac{r^2}{2^{8n_x}} \right)^{\frac{1}{2}} BA + \left( \frac{r^2(k_y r^2 + 2^{n_x})}{2^{8n_x}(k_y r^2)} \right)^{\frac{1}{2}} CA - \left( \frac{r^2(k_y r^2 + 2^{n_x})}{2^{8n_x}(k_y r^2)} \right)^{\frac{1}{2}} BA \\
&\quad + \left( \frac{k_y r^4}{2^{4n_x}(k_y r^2 + 2^{n_x})} \right)^{\frac{1}{2}} B - \left( \frac{k_y r^4}{2^{8n_x}(k_y r^2 + 2^{n_x})} \right)^{\frac{1}{2}} BA
\end{aligned}
$$

$$+ \left(\frac{r^2}{2^{4n_x}}\right)^{\frac{1}{2}} C(I - \frac{1}{2^{2n_x}}A) - \left(\frac{r^2}{2^{4n_x}}\right)^{\frac{1}{2}} B(I - \frac{1}{2^{2n_x}}A)$$

$$= \left(\frac{r^2}{2^{4n_x}}\right)^{\frac{1}{2}} B + \left(\frac{r^2(k_y r^2 + 2^{n_x})}{2^{4n_x}(k_y r^2)}\right)^{\frac{1}{2}} B - \left(\frac{r^2(k_y r^2 + 2^{n_x})}{2^{4n_x}(k_y r^2)}\right)^{\frac{1}{2}} B$$

$$+ \left(\frac{k_y r^4}{2^{4n_x}(k_y r^2 + 2^{n_x})}\right)^{\frac{1}{2}} B - \left(\frac{k_y r^4}{2^{4n_x}(k_y r^2 + 2^{n_x})}\right)^{\frac{1}{2}} B$$

$$+ \left(\frac{r^2}{2^{4n_x}}\right)^{\frac{1}{2}} C(I - \frac{1}{2^{2n_x}}A) - \left(\frac{r^2}{2^{4n_x}}\right)^{\frac{1}{2}} B(I - \frac{1}{2^{2n_x}}A)$$

$$= \left(\frac{r^2}{2^{4n_x}}\right)^{\frac{1}{2}} B + \left(\frac{r^2}{2^{4n_x}}\right)^{\frac{1}{2}} C - \left(\frac{r^2}{2^{4n_x}}\right)^{\frac{1}{2}} CA$$

$$- \left(\frac{r^2}{2^{4n_x}}\right)^{\frac{1}{2}} B + \left(\frac{r^2}{2^{4n_x}}\right)^{\frac{1}{2}} BA$$

$$= \left(\frac{r^2}{2^{4n_x}}\right)^{\frac{1}{2}} C - rB + rB$$

$$= \frac{r}{2^{2n_x}} \Omega_x^T \Omega_x \Omega_x^T = \frac{r}{2^{n_x}} \Omega_x^T$$

$$Q_4 = \left[\left(\frac{r^2}{2^{3n_x}(k_y r^2 + 2^{n_x})}\right)^{\frac{1}{2}} B + \left(\frac{r^2}{2^{3n_x}(k_y r^2)}\right)^{\frac{1}{2}} (C - B)\right]$$

$$\left[\left(\frac{r^2(k_y r^2 + 2^{n_x})}{2^{7n_x}}\right)^{\frac{1}{2}} A\Omega_x + \left(\frac{k_y r^4}{2^{3n_x}}\right)^{\frac{1}{2}} (I - \frac{1}{2^{2n_x}}A)\Omega_x\right] + T\frac{r^2}{2^{n_x}}P^T$$

$$= \left[\left(\frac{r^2}{2^{3n_x}(k_y r^2 + 2^{n_x})}\right)^{\frac{1}{2}} B + \left(\frac{r^2}{2^{3n_x}(k_y r^2)}\right)^{\frac{1}{2}} (C - B)\right] \left(\frac{r^2(k_y r^2 + 2^{n_x})}{2^{7n_x}}\right)^{\frac{1}{2}} A\Omega_x$$

$$+ \left[\left(\frac{r^2}{2^{3n_x}(k_y r^2 + 2^{n_x})}\right)^{\frac{1}{2}} B + \left(\frac{r^2}{2^{3n_x}(k_y r^2)}\right)^{\frac{1}{2}} (C - B)\right] \left(\frac{k_y r^4}{2^{3n_x}}\right)^{\frac{1}{2}} (I - \frac{1}{2^{2n_x}}A)\Omega_x$$

$$+ T\frac{r^2}{2^{n_x}}P^T$$

$$= \left(\frac{r^4}{2^{10n_x}}\right)^{\frac{1}{2}} BA\Omega_x + \left(\frac{r^4(k_y r^2 + 2^{n_x})}{2^{10n_x}(k_y r^2)}\right)^{\frac{1}{2}} (C - B)A\Omega_x$$

$$+ \left(\frac{k_y r^6}{2^{6n_x}(k_y r^2 + 2^{n_x})}\right)^{\frac{1}{2}} B(I - \frac{1}{2^{2n_x}}A)\Omega_x$$

$$+ \left(\frac{r^4}{2^{6n_x}}\right)^{\frac{1}{2}} (C - B)(I - \frac{1}{2^{2n_x}}A)\Omega_x + T\frac{r^2}{2^{n_x}}P^T$$

$$= \left(\frac{r^4}{2^{10n_x}}\right)^{\frac{1}{2}} BA\Omega_x + \left(\frac{r^4(k_y r^2 + 2^{n_x})}{2^{10n_x}(k_y r^2)}\right)^{\frac{1}{2}} CA\Omega_x - \left(\frac{r^4(k_y r^2 + 2^{n_x})}{2^{10n_x}(k_y r^2)}\right)^{\frac{1}{2}} BA\Omega_x$$

$$+ \left(\frac{k_y r^6}{2^{6n_x}(k_y r^2 + 2^{n_x})}\right)^{\frac{1}{2}} B\Omega_x - \left(\frac{k_y r^6}{2^{10n_x}(k_y r^2 + 2^{n_x})}\right)^{\frac{1}{2}} BA\Omega_x$$

$$+ \left(\frac{r^4}{2^{6n_x}}\right)^{\frac{1}{2}} C(I - \frac{1}{2^{2n_x}}A)\Omega_x - \left(\frac{r^4}{2^{6n_x}}\right)^{\frac{1}{2}} B(I - \frac{1}{2^{2n_x}}A)\Omega_x + T\frac{r^2}{2^{n_x}}P^T$$

$$= \left(\frac{r^4}{2^{6n_x}}\right)^{\frac{1}{2}} B\Omega_x + \left(\frac{r^4(k_y r^2 + 2^{n_x})}{2^{6n_x}(k_y r^2)}\right)^{\frac{1}{2}} B\Omega_x - \left(\frac{r^4(k_y r^2 + 2^{n_x})}{2^{6n_x}(k_y r^2)}\right)^{\frac{1}{2}} B\Omega_x$$

$$+ \left( \frac{k_y r^6}{2^{6n_x}(k_y r^2 + 2^{n_x})} \right)^{\frac{1}{2}} B\Omega_x - \left( \frac{k_y r^6}{2^{6n_x}(k_y r^2 + 2^{n_x})} \right)^{\frac{1}{2}} B\Omega_x$$

$$+ \left( \frac{r^4}{2^{6n_x}} \right)^{\frac{1}{2}} C(I - \frac{1}{2^{2n_x}}A)\Omega_x - \left( \frac{r^4}{2^{6n_x}} \right)^{\frac{1}{2}} B(I - \frac{1}{2^{2n_x}}A)\Omega_x + T\frac{r^2}{2^{n_x}}P^T$$

$$= \left( \frac{r^4}{2^{6n_x}} \right)^{\frac{1}{2}} B\Omega_x + \left( \frac{r^4}{2^{6n_x}} \right)^{\frac{1}{2}} C\Omega_x - \left( \frac{r^4}{2^{6n_x}} \right)^{\frac{1}{2}} \frac{1}{2^{2n_x}}CA\Omega_x$$

$$- \left( \frac{r^4}{2^{6n_x}} \right)^{\frac{1}{2}} B\Omega_x + \left( \frac{r^4}{2^{6n_x}} \right)^{\frac{1}{2}} \frac{1}{2^{2n_x}}BA\Omega_x + T\frac{r^2}{2^{n_x}}P^T$$

$$= \left( \frac{r^4}{2^{6n_x}} \right)^{\frac{1}{2}} B\Omega_x + \left( \frac{r^4}{2^{6n_x}} \right)^{\frac{1}{2}} C\Omega_x - \left( \frac{r^4}{2^{6n_x}} \right)^{\frac{1}{2}} B\Omega_x$$

$$- \left( \frac{r^4}{2^{6n_x}} \right)^{\frac{1}{2}} B\Omega_x + \left( \frac{r^4}{2^{6n_x}} \right)^{\frac{1}{2}} B\Omega_x + T\frac{r^2}{2^{n_x}}P^T$$

$$= \frac{r^2}{2^{3n_x}}C\Omega_x + T\frac{r^2}{2^{n_x}}P^T$$

$$= \frac{r^2}{2^{3n_x}}\Omega_x^T\Omega_x\Omega_x^T\Omega_x + T\frac{r^2}{2^{n_x}}P^T$$

$$= \frac{r^2}{2^{2n_x}}\Omega_x^T\Omega_x + T\frac{r^2}{2^{n_x}}P^T$$

$$= \frac{r^2}{2^{2n_x}}\Omega_x^T\Omega_x + \frac{r^2(k_x k_y)}{2^{n_x}}T(\frac{1}{k_x k_y})^{\frac{1}{2}}P^T$$

$$= \frac{r^2}{2^{2n_x}}\Omega_x^T\Omega_x + \frac{r^2(k_x k_y)^{\frac{1}{2}}}{2^{n_x}}(\frac{1}{k_x k_y})^{\frac{1}{2}}\mathbf{I_{2^{n_x} \times 2^{n_x}}} - \frac{r^2(k_x k_y)^{\frac{1}{2}}}{2^{n_x}}(\frac{1}{k_x k_y})^{\frac{1}{2}}(\frac{1}{2^{n_x}})\Omega_x^T\Omega_x$$

$$= \frac{r^2}{2^{2n_x}}\Omega_x^T\Omega_x + \frac{r^2}{2^{n_x}}\mathbf{I_{2^{n_x} \times 2^{n_x}}} - \frac{r^2}{2^{2n_x}}\Omega_x^T\Omega_x = \frac{r^2}{2^{n_x}}\mathbf{I_{2^{n_x} \times 2^{n_x}}}$$

Thus the final SVD of $\Sigma^{yx}$ is (note that quadrants 2,3 and 4 repeat based on the values of $k_x$ and $k_y$):

$$USV^T = \begin{bmatrix} \frac{1}{2^{n_x}}\Omega_y\Omega_x^T & \frac{r}{2^{n_x}}\Omega_y & \cdots & \frac{r}{2^{n_x}}\Omega_y \\ \vdots & \vdots & \ddots & \\ \frac{r}{2^{n_x}}\Omega_x^T & \frac{r^2}{2^{n_x}}\mathbf{I_{2^{n_x} \times 2^{n_x}}} & \cdots & \frac{r^2}{2^{n_x}}\mathbf{I_{2^{n_x} \times 2^{n_x}}} \end{bmatrix} = \frac{1}{2^{n_x}}YX^T$$

We now show that the $V$ and $U$ matrices are orthogonal:

$$V^T V = \begin{bmatrix} \left( \frac{2^{n_x}}{(k_x r^2 + 2^{n_x})} \right)^{\frac{1}{2}}\mathbf{I_{n_x \times n_x}} & \left( \frac{r^2}{2^{n_x}(k_x r^2 + 2^{n_x})} \right)^{\frac{1}{2}}\Omega_x \\ \mathbf{0_{k_x 2^{n_x} \times n_x}} & (\frac{1}{k_x})^{\frac{1}{2}}P^T \end{bmatrix} \begin{bmatrix} \left( \frac{2^{n_x}}{(k_x r^2 + 2^{n_x})} \right)^{\frac{1}{2}}\mathbf{I_{n_x \times n_x}} & \mathbf{0_{n_x \times k_x 2^{n_x}}} \\ \left( \frac{r^2}{2^{n_x}(k_x r^2 + 2^{n_x})} \right)^{\frac{1}{2}}\Omega_x^T & (\frac{1}{k_x})^{\frac{1}{2}}P \end{bmatrix}$$

$$= \begin{bmatrix} \frac{2^{n_x}}{(k_x r^2 + 2^{n_x})}\mathbf{I_{n_x \times n_x}} + k_x\left( \frac{r^2}{2^{n_x}(k_x r^2 + 2^{n_x})} \right)\Omega_x\Omega_x^T & \left( \frac{r^2}{2^{n_x}(k_x r^2 + 2^{n_x})(k_x)} \right)^{\frac{1}{2}}\Omega_x P \\ \left( \frac{r^2}{2^{n_x}(k_x r^2 + 2^{n_x})(k_x)} \right)^{\frac{1}{2}}P^T\Omega_x^T & (k_x)(\frac{1}{k_x})P^T P \end{bmatrix}$$

$$= \begin{bmatrix} \frac{2^{n_x}}{(k_x r^2 + 2^{n_x})}\mathbf{I_{n_x \times n_x}} + \left( \frac{k_x r^2}{(k_x r^2 + 2^{n_x})} \right)\mathbf{I_{n_x \times n_x}} & \left( \frac{r^2}{2^{n_x}(k_x r^2 + 2^{n_x})(k_x)} \right)^{\frac{1}{2}}\Omega_x P \\ \left( \frac{r^2}{2^{n_x}(k_x r^2 + 2^{n_x})(k_x)} \right)^{\frac{1}{2}}P^T\Omega_x^T & \mathbf{I_{2^{n_x} \times 2^{n_x}}} \end{bmatrix}$$

$$= \begin{bmatrix} \mathbf{I_{n_x \times n_x}} & \mathbf{0_{n_x \times 2^{n_x}}} \\ \mathbf{0_{2^{n_x} \times n_x}} & \mathbf{I_{2^{n_x} \times 2^{n_x}}} \end{bmatrix}$$

$$U^T U = \begin{bmatrix} \left(\frac{1}{2^{n_x}(k_y r^2 + 2^{n_x})}\right)^{\frac{1}{2}} \Omega_x \Omega_y^T & \left(\frac{r^2}{2^{3n_x}(k_y r^2 + 2^{n_x})}\right)^{\frac{1}{2}} B^T + \left(\frac{r^2}{2^{3n_x}(k_y r^2)}\right)^{\frac{1}{2}} (C-B)^T \\ \mathbf{0}_{\mathbf{k_y 2^{n_x} \times n_y}} & (\frac{1}{k_y})^{\frac{1}{2}} T^T \end{bmatrix}$$

$$\begin{bmatrix} \left(\frac{1}{2^{n_x}(k_y r^2 + 2^{n_x})}\right)^{\frac{1}{2}} \Omega_y \Omega_x^T & \mathbf{0}_{\mathbf{n_y \times k_x 2^{n_x}}} \\ \left(\frac{r^2}{2^{3n_x}(k_y r^2 + 2^{n_x})}\right)^{\frac{1}{2}} B + \left(\frac{r^2}{2^{3n_x}(k_y r^2)}\right)^{\frac{1}{2}} (C-B) & (\frac{1}{k_y})^{\frac{1}{2}} T \end{bmatrix}$$

We first consider Quadrant 1:

$$Q_1 = \left(\frac{1}{2^{n_x}(k_y r^2 + 2^{n_x})}\right) \Omega_x \Omega_y^T \Omega_y \Omega_x^T$$

$$+ k_y \left[\left(\frac{r^2}{2^{3n_x}(k_y r^2 + 2^{n_x})}\right)^{\frac{1}{2}} B^T + \left(\frac{r^2}{2^{3n_x}(k_y r^2)}\right)^{\frac{1}{2}} (C-B)^T\right]$$

$$\left[\left(\frac{r^2}{2^{3n_x}(k_y r^2 + 2^{n_x})}\right)^{\frac{1}{2}} B + \left(\frac{r^2}{2^{3n_x}(k_y r^2)}\right)^{\frac{1}{2}} (C-B)\right]$$

$$= \left(\frac{1}{2^{n_x}(k_y r^2 + 2^{n_x})}\right) \Omega_x \Omega_y^T \Omega_y \Omega_x^T$$

$$+ k_y \left[\left(\frac{r^2}{2^{3n_x}(k_y r^2 + 2^{n_x})}\right)^{\frac{1}{2}} B^T + \left(\frac{r^2}{2^{3n_x}(k_y r^2)}\right)^{\frac{1}{2}} (C-B)^T\right] \left(\frac{r^2}{2^{3n_x}(k_y r^2 + 2^{n_x})}\right)^{\frac{1}{2}} B$$

$$+ k_y \left[\left(\frac{r^2}{2^{3n_x}(k_y r^2 + 2^{n_x})}\right)^{\frac{1}{2}} B^T + \left(\frac{r^2}{2^{3n_x}(k_y r^2)}\right)^{\frac{1}{2}} (C-B)^T\right] \left(\frac{r^2}{2^{3n_x}(k_y r^2)}\right)^{\frac{1}{2}} (C-B)$$

$$= \left(\frac{1}{2^{n_x}(k_y r^2 + 2^{n_x})}\right) \Omega_x \Omega_y^T \Omega_y \Omega_x^T$$

$$+ k_y \frac{r^2}{2^{3n_x}(k_y r^2 + 2^{n_x})} B^T B + k_y \left(\frac{r^4}{2^{6n_x}(k_y r^2)(k_y k_y r^2 + 2^{n_x})}\right)^{\frac{1}{2}} (C-B)^T B$$

$$+ k_y \left(\frac{r^4}{2^{6n_x}(k_y r^2)(k_y r^2 + 2^{n_x})}\right)^{\frac{1}{2}} B^T (C-B)$$

$$+ k_y \left(\frac{r^2}{2^{3n_x}(k_y r^2)}\right) (C-B)^T (C-B)$$

It is helpful to note that $(C-B)^T B = \mathbf{0}_{\mathbf{n_x \times n_x}}$ and $\Omega_x \Omega_y^T \Omega_y \Omega_x^T = (\frac{1}{2^{n_x}}) B^T B$

$$Q_1 = \left(\frac{2^{n_x}}{2^{3n_x}(k_y r^2 + 2^{n_x})}\right) B^T B + \frac{k_y r^2}{2^{3n_x}(k_y r^2 + 2^{n_x})} B^T B$$

$$+ \left(\frac{k_y r^2}{2^{3n_x}(k_y r^2)}\right) (C-B)^T (C-B)$$

$$Q_1 = \left(\frac{k_y r^2 + 2^{n_x}}{2^{3n_x}(k_y r^2 + 2^{n_x})}\right) B^T B + \left(\frac{1}{2^{3n_x}}\right) (C-B)^T (C-B)$$

$$Q_1 = \left(\frac{\mathbf{1}}{2^{3n_x}}\right) B^T B + \left(\frac{1}{2^{3n_x}}\right) (C-B)^T (C-B)$$

$$Q_1 = \mathbf{I_{n_x \times n_x}}$$

Where we used the fact that $B^T B + (C-B)^T (C-B) = 2^{3n_x} \mathbf{I_{n_x \times n_x}}$.
Thus:

$$U^T U = \begin{bmatrix} \left(\frac{1}{2^{n_x}(k_y r^2 + 2^{n_x})}\right)^{\frac{1}{2}} \Omega_x \Omega_y^T & \left(\frac{r^2}{2^{3n_x}(k_y r^2 + 2^{n_x})}\right)^{\frac{1}{2}} B^T + \left(\frac{r^2}{2^{3n_x}(k_y r^2)}\right)^{\frac{1}{2}} (C-B)^T \\ \mathbf{0}_{\mathbf{k_x 2^{n_x} \times n_y}} & (\frac{1}{k_y})^{\frac{1}{2}} T^T \end{bmatrix}$$

$$\begin{bmatrix} \left(\frac{1}{2^{n_x}(k_y r^2 + 2^{n_x})}\right)^{\frac{1}{2}} \Omega_y \Omega_x^T & \mathbf{0_{n_y \times k_x 2^{n_x}}} \\ \left(\frac{r^2}{2^{3n_x}(k_y r^2 + 2^{n_x})}\right)^{\frac{1}{2}} B + \left(\frac{r^2}{2^{3n_x}(k_y r^2)}\right)^{\frac{1}{2}} (C - B) & (\frac{1}{k_y})^{\frac{1}{2}} T \end{bmatrix}$$

$$= \begin{bmatrix} \mathbf{I_{n_x \times n_x}} & \left(\frac{k_y r^2}{2^{3n_x}(k_y r^2 + 2^{n_x})}\right)^{\frac{1}{2}} B^T T + \left(\frac{1}{2^{3n_x}}\right)^{\frac{1}{2}} (C - B)^T T \\ \left(\frac{k_y r^2}{2^{3n_x}(k_y^2 r^2 + 2^{n_x})}\right)^{\frac{1}{2}} T^T B + \left(\frac{1}{2^{3n_x}}\right)^{\frac{1}{2}} T^T (C - B) & k_y(\frac{1}{k_y}) T^T T \end{bmatrix}$$

$$= \begin{bmatrix} \mathbf{I_{n_x \times n_x}} & \left(\frac{k_y r^2}{2^{3n_x}(k_y r^2 + 2^{n_x})}\right)^{\frac{1}{2}} B^T T + \left(\frac{1}{2^{3n_x}}\right)^{\frac{1}{2}} (C - B)^T T \\ \left(\frac{k_y r^2}{2^{3n_x}(k_y^2 r^2 + 2^{n_x})}\right)^{\frac{1}{2}} T^T B + \left(\frac{1}{2^{3n_x}}\right)^{\frac{1}{2}} T^T (C - B) & k_y(\frac{1}{k_y}) T^T T \end{bmatrix}$$

$$= \begin{bmatrix} \mathbf{I_{n_x \times n_x}} & \mathbf{0_{n_x \times 2^{n_x}}} \\ \mathbf{0_{2^{n_x} \times n_x}} & \mathbf{I_{2^{n_x} \times 2^{n_x}}} \end{bmatrix}$$

Thus, the we have proven that $USV^T = \Sigma^{yx}$ and that the singular vector matrices are orthogonal. Since the SVD of the $\Sigma^x$ matrix is a special case of $\Sigma^{yx}$ with $n_y = n_x$ then this also proves that the SVD for $\Sigma^x = VDV^T$ with orthogonal singular vector matrices.

## E    INPUT AND OUTPUT PARTITIONED FROBENIUS NORMS

We partition the input-output mapping along the compositional and non-compositional input and output components. Figure 11 shows this partitioning and how it relates to the SVD of the covariance matrix, while the time-courses for these norms can be seen in Equations 17 to 20. From these equations we see that the mappings to both components of the output rely on all inputs. Thus, the non-compositional inputs still offer some benefit to the compositional output and are used in the network mapping. Likewise the compositional inputs are used for the non-compositional mapping.

$$\Omega_x \Omega_y\text{-Norm} = \left( \frac{2^{2n_x} n_y}{(k_x r^2 + 2^{n_x})(k_y r^2 + 2^{n_x})} \pi_1^2(t) \right)^{\frac{1}{2}} \tag{17}$$

$$\Gamma_x \Omega_y\text{-Norm} = \left( \frac{2^{n_x} n_y k_x r^2}{(k_x r^2 + 2^{n_x})(k_y r^2 + 2^{n_x})} \pi_1^2(t) \right)^{\frac{1}{2}} \tag{18}$$

$$\Omega_x \Gamma_y\text{-Norm} = \left( \frac{2^{n_x} k_y n_y r^2}{(k_x r^2 + 2^{n_x})(k_y r^2 + 2^{n_x})} \pi_1^2(t) + \frac{2^{n_x}(n_x - n_y)}{k_x r^2 + 2^{n_x}} \pi_2^2(t) \right)^{\frac{1}{2}} \tag{19}$$

$$\Gamma_x \Gamma_y\text{-Norm} = \left( \frac{k_x k_y n_y r^4}{(k_x r^2 + 2^{n_x})(k_y r^2 + 2^{n_x})} \pi_1^2(t) + \frac{(n_x - n_y) k_x r^2}{k_x r^2 + 2^{n_x}} \pi_2^2(t) + (2^{n_x} - n_x) \pi_3^2(t) \right)^{\frac{1}{2}} \tag{20}$$

Figure 12 shows the simulated and predicted training dynamics for these norms on a deep linear network. We note that the mapping to the compositional output relies evenly on both the compositional and non-compositional inputs, with a slight preference to the compositional inputs. Similarly, the mapping to the non-compositional outputs also uses the compositional and non-compositional inputs evenly, with a preference towards the non-compositional inputs towards the end of training. In Appendix F we consider the same norm equations for the split network architectures.

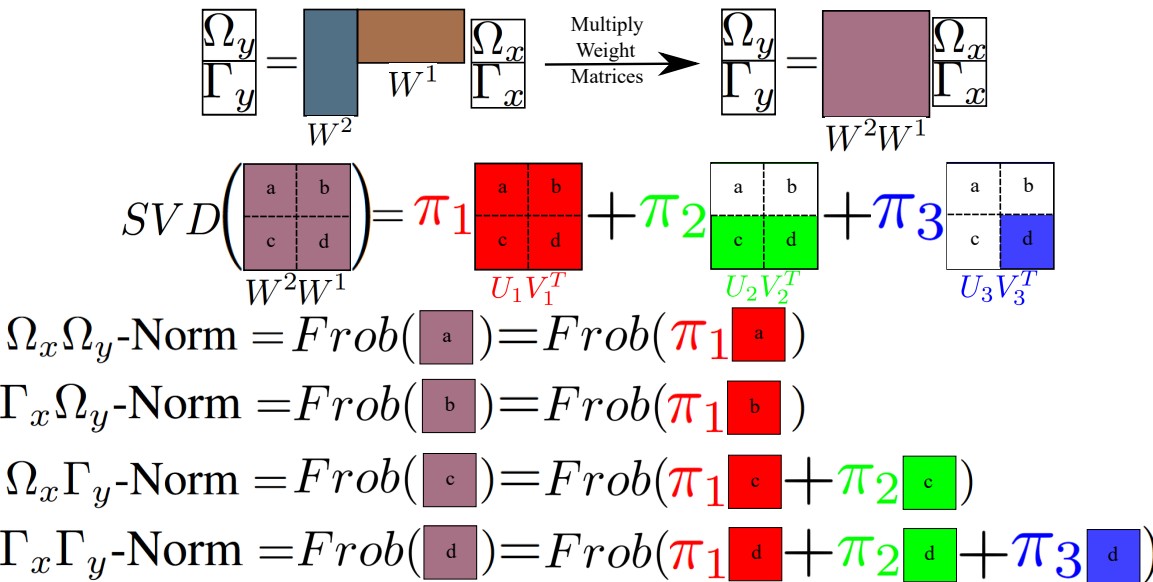

Figure 11: Representation of the partitioned norms and their relation to the SVD of the covariance matrix. Specifically, we consider the norm of pieces of the network mapping individually, but these pieces only depend on certain modes of variation. By analysing the time-scale of learning and final value of the norms we can determine the network's ability to systematically generalise.

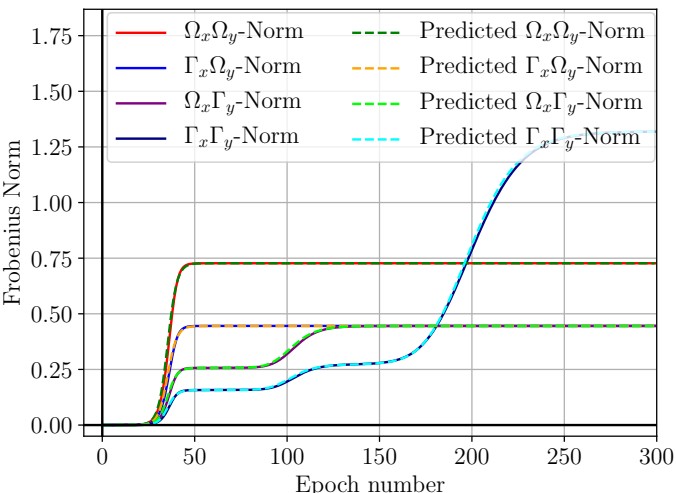

Figure 12: Frobenius norms of the deep network mapping partitioned by the compositional and non-compositional inputs and output on a dense linear neural module.

## F MODULARITY AND ARCHITECTURE

In this Section we now consider the norm equations for the modular network architectures. In Figure 13b we depict the Frobenius norms of the mapping for the output-partitioned network (the network's Singular Values are shown in Figure 13a). The dynamics of this case can equally be seen as using one neural module responsible for the compositional output ($\Omega_y$) mapping and another for the non-compositional output ($\Gamma_y$) mapping. From the perspective of the $\Gamma_y$ mapping the compositional output features ($\Omega_y$) are not present ($n_y$ is effectively set to 0 for this module), and the mapping will no longer be dependent on $\pi_1$. The compositional output mapping is still only dependent on the $\pi_1$ effective singular value. The time-courses for the resultant Frobenius norms for the module

mapping to the compositional output are given in Equations 21 and 22 (by substituting $k_y = 0$ in the norm Equation from Appendix E). Similarly, the time-courses for the resultant Frobenius norms for the module mapping to the non-compositional output are given in Equations 23 and 24 (by substituting $n_y = 0$ in the norm Equation from Appendix E). From these equations we see that the two mappings use entirely different modes of variation, and as a result the learning of one mapping does not imply any progress in the learning of the other. However, even with the output partitioning, the norm connecting non-compositional input to compositional output is not 0. Nor is the norm for the compositional input to non-compositional output mapping. Thus the output partitioned network will still not learn a systematic mapping for all datasets in the space.

$$\Omega_x \Omega_y\text{-Norm} = \left( \frac{2^{n_x} n_y}{k_x r^2 + 2^{n_x}} \pi_1^2(t) \right)^{\frac{1}{2}} \tag{21}$$

$$\Gamma_x \Omega_y\text{-Norm} = \left( \frac{n_y k_x r^2}{k_x r^2 + 2^{n_x}} \pi_1^2(t) \right)^{\frac{1}{2}} \tag{22}$$

$$\Omega_x \Gamma_y\text{-Norm} = \left( \frac{2^{n_x} n_x}{k_x r^2 + 2^{n_x}} \pi_2^2(t) \right)^{\frac{1}{2}} \tag{23}$$

$$\Gamma_x \Gamma_y\text{-Norm} = \left( \frac{n_x k_x r^2}{k_x r^2 + 2^{n_x}} \pi_2^2(t) + (2^{n_x} - n_x) \pi_3^2(t) \right)^{\frac{1}{2}} \tag{24}$$

Finally, we consider the fully partitioned network. In this case there are two modules, one connecting compositional input and output components and another connecting the non-compositional input and output components. The module connecting the compositional components is now restricted to working with effective datasets with the hyper-parameters of $k_x = k_y = 0$. Substituting this into the norm equations from Appendix E produces a single norm shown in Equation 25. Similarly, the non-compositional module trains on effective datasets with $n_x = n_y = 0$ and has a single norm shown in Equation 26. We note that this, very strict architectural bias of a perfect partitioning of compositional and non-compositional components, is enough for the network to systematically generalize, since no non-compositional components affect mappings to/from compositional components. Thus, only the most perfect modularity is able to achieve systematicity as defined by Definition 3.1. We consider the case of imperfect output partitions in Appendix G and find the network will no longer be systematic.

$$\Omega_x \Omega_y\text{-Norm} = \sqrt{n_y} \pi_1(t) \tag{25}$$

$$\Gamma_x \Gamma_y\text{-Norm} = \pi_3(t) \tag{26}$$

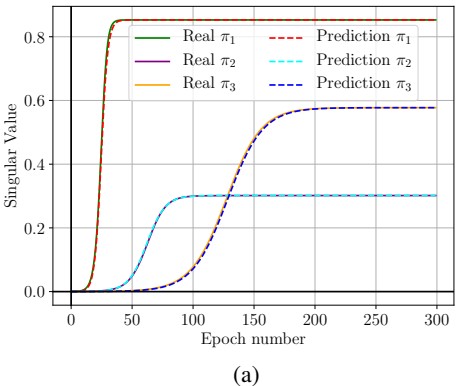

(a)

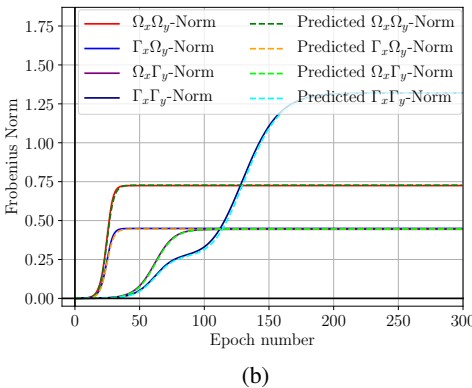

(b)

Figure 13: Training dynamics of the deep network mapping partitioned by the compositional and non-compositional inputs and output on two linear neural modules. One module maps exclusively to the compositional output, while the other maps exclusively to the non-compositional output. a) Learning trajectories of the three unique Singular Values learned by the output-split network. b) Frobenius norms of the output partitioned network.

## G   IMPERFECT OUTPUT PARTITIONS

We now investigate the case where the split network architecture does not perfectly partition the output into the compositional and non-compositional components. In this case some of the non-compositional identity output blocks are grouped with the compositional outputs (we only consider partitioning along the compositional and non-compositional blocks to keep the closed form solutions tractable). Thus, we separate the number of non-compositional outputs $k_y$ into the number of non-compositional outputs of the left network branch $k_y^{\text{left}}$ and right network branch $k_y^{\text{right}}$, such that $k_y^{\text{left}} + k_y^{\text{right}} = k_y$.

Equations 27 to 32 show the set of norms which emerge. In the extreme cases when $k_y^{\text{left}} = k_y$ we recover the dense network equations shown in Equations 17 to 20, since $\pi_2 = \pi_3 = 0$ for Equations 31 and 32. This is apparent from the singular value equations for $\pi_2$ and $\pi_3$ shown in Equations 7 and 9 with $k_y$ in the numerator which is replaced by $k_y^{\text{right}} = 0$ for the right module. Thus, Equation 31 and 32 fall away completely. In the other extreme case of $k_y^{\text{right}} = k_y$ we recover the split network equations shown in Equations 21 to 24. This is again because $\pi_2 = \pi_3 = 0$ but this time from the left network branch's perspective. By definition $k_y^{\text{left}} = 0$ in this case and so all components of Equations 29 and 30 fall away, leaving just Equations 27 and 28 with Equations 31 and 32. The most important conclusion to be drawn, however, is that if any non-compositional components are partitioned with compositional components then the module will behave similarly to the dense network of Section 5. This can be shown the same way as in the proof for Observation 5.1, by noting that Equations 29 and 30 will be non-zero for any case where the compositional input to compositional output is learned (due to a shared mode of variation). Thus, only the most stringent and perfectly allocated network modules will be able to specialize.

$$\Omega_x \Omega_y\text{-Norm} = \left( \frac{2^{2n_x} n_y}{(k_x r^2 + 2^{n_x})(k_y^{left} r^2 + 2^{n_x})} \pi_1^2(t) \right)^{\frac{1}{2}} \tag{27}$$

$$\Gamma_x \Omega_y\text{-Norm} = \left( \frac{2^{n_x} n_y k_x r^2}{(k_x r^2 + 2^{n_x})(k_y^{left} r^2 + 2^{n_x})} \pi_1^2(t) \right)^{\frac{1}{2}} \tag{28}$$

$$\Omega_x \Gamma_y^{left}\text{-Norm} = \left( \frac{2^{n_x} k_y^{left} n_y r^2}{(k_x r^2 + 2^{n_x})(k_y^{left} r^2 + 2^{n_x})} \pi_1^2(t) + \frac{2^{n_x}(n_x - n_y)}{k_x r^2 + 2^{n_x}} \pi_2^2(t) \right)^{\frac{1}{2}} \tag{29}$$

$$\Gamma_x \Gamma_y^{left}\text{-Norm} = \left( \frac{k_x k_y^{left} n_y r^4}{(k_x r^2 + 2^{n_x})(k_y^{left} r^2 + 2^{n_x})} \pi_1^2(t) + \frac{(n_x - n_y) k_x r^2}{k_x r^2 + 2^{n_x}} \pi_2^2(t) + (2^{n_x} - n_x)\pi_3^2(t) \right)^{\frac{1}{2}} \tag{30}$$

$$\Omega_x \Gamma_y^{right}\text{-Norm} = \left( \frac{2^{n_x} n_x}{k_x r^2 + 2^{n_x}} \pi_2^2(t) \right)^{\frac{1}{2}} \tag{31}$$

$$\Gamma_x \Gamma_y^{right}\text{-Norm} = \left( \frac{n_x k_x r^2}{k_x r^2 + 2^{n_x}} \pi_2^2(t) + (2^{n_x} - n_x)\pi_3^2(t) \right)^{\frac{1}{2}} \tag{32}$$

## H   PROOFS OF OBSERVATIONS

### H.1   PROOF OF OBSERVATION 5.1

In this section we prove **Observation 5.1.**: *For all datasets in the space it is impossible for a dense architecture to learn a systematic mapping between compositional components.*

**Proof:** To prove this we are required to show that: For all points in the space of datasets: $n_x, n_y, k_x, k_y, r \in \mathbb{Z}^+$ that $\Omega_x \Omega_y$-Norm$> 0 \Rightarrow \Gamma_x \Omega_y$-Norm$> 0$ and $\Omega_x \Gamma_y$-Norm$> 0$ for $t \in \mathbb{Z}^+$.

Thus, we assume that $\Omega_x\Omega_y$-Norm$> 0$ and use the fact that singular values are positive semi-definite to show that:

$$\Omega_x\Omega_y\text{-Norm}^2 = \frac{2^{2n_x}n_y}{(k_xr^2 + 2^{n_x})(k_yr^2 + 2^{n_x})}\pi_1^2(t)$$
$$= a\pi_1^2(t)$$
$$> 0$$

since $a = \frac{2^{2n_x}n_y}{(k_xr^2 + 2^{n_x})(k_yr^2 + 2^{n_x})} > 0$. Given that $\pi_1^2(t) > 0$ and $n_x, n_y, k_x, k_y, r \in \mathbb{Z}^+$ then for the $\Gamma_x\Omega_y$ and $\Omega_x\Gamma_y$-Norms:

$$\Gamma_x\Omega_y\text{-Norm}^2 = \frac{2^{n_x}n_yk_xr^2}{(k_xr^2 + 2^{n_x})(k_yr^2 + 2^{n_x})}\pi_1^2(t)$$
$$= b\pi_1^2(t)$$
$$> 0$$
$$\to \Gamma_x\Omega_y\text{-Norm} > 0.$$

and

$$\Omega_x\Gamma_y\text{-Norm}^2 = \frac{2^{n_x}k_yn_yr^2}{(k_xr^2 + 2^{n_x})(k_yr^2 + 2^{n_x})}\pi_1^2(t) + \frac{2^{n_x}(n_x - n_y)}{k_xr^2 + 2^{n_x}}\pi_2^2(t)$$
$$> \frac{2^{n_x}k_yn_yr^2}{(k_xr^2 + 2^{n_x})(k_yr^2 + 2^{n_x})}\pi_1^2(t)$$
$$= c\pi_1^2(t)$$
$$> 0$$
$$\to \Omega_x\Gamma_y\text{-Norm} > 0$$

where $b = \frac{2^{n_x}n_yk_xr^2}{(k_xr^2+2^{n_x})(k_yr^2+2^{n_x})} > 0$ and $c = \frac{2^{n_x}k_yn_yr^2}{(k_xr^2+2^{n_x})(k_yr^2+2^{n_x})} > 0$

### H.2 Proof of Observation 6.1

In this section we prove **Observation 6.1.**: *Modularity does impose some form of bias towards systematicity but if any non-compositional structure is considered in the input then a systematic mapping will not be learned.*

**Proof:** To prove this we are required to show that: For all points in the space of datasets with the output partitioned network: $n_x, n_y, k_x, k_y, r \in \mathbb{Z}^+$ that $\Omega_x\Omega_y$-Norm $> 0 \not\Rightarrow \Omega_x\Gamma_y$-Norm $> 0$ for $t \in \mathbb{Z}^+$ and $\Omega_x\Omega_y$-Norm $> 0 \Rightarrow \Gamma_x\Omega_y$-Norm $> 0$ for $t \in \mathbb{Z}^+$. This shows that, while modularity removes the guaranteed association between compositional input and non-compositional output it is still impossible to remove the association of non-compositional input and compositional output.

We assume that $\Omega_x\Omega_y$-Norm$> 0$ and use that the singular values are positive semi-definite to show:

$$\Omega_x\Omega_y\text{-Norm}^2 = \frac{2^{n_x}n_y}{k_xr^2 + 2^{n_x}}\pi_1^2(t)$$
$$= a\pi_1^2(t)$$
$$> 0$$

since $a = \frac{2^{n_x}n_y}{k_xr^2 + 2^{n_x}} > 0$. Given that $\pi_1^2(t) > 0$ and $n_x, n_y, k_x, k_y, r \in \mathbb{Z}^+$ then for the $\Gamma_x\Omega_y$-Norm:

$$\Gamma_x\Omega_y\text{-Norm}^2 = \frac{n_yk_xr^2}{k_xr^2 + 2^{n_x}}\pi_1^2(t)$$
$$= b\pi_1^2(t)$$
$$> 0$$
$$\to \Gamma_x\Omega_y\text{-Norm} > 0$$

where $b = \frac{n_yk_xr^2}{k_xr^2+2^{n_x}} > 0$.

However for the $\Omega_x \Gamma_y$-Norm:

$$\begin{aligned}
\Omega_x \Gamma_y\text{-Norm}^2 &= \frac{2^{n_x} n_x}{k_x r^2 + 2^{n_x}} \pi_2^2(t) \\
&= c \pi_2^2(t)
\end{aligned}$$

The value of $\pi_2^2(t)$ is not certain at time $t$ just from knowing that $\Omega_x \Omega_y$-Norm $> 0$. Thus, $\Omega_x \Omega_y$-Norm $> 0 \not\Rightarrow \Omega_x \Gamma_y$-Norm $> 0$ for $t \in \mathbb{Z}^+$. However, it is still the case that $\Omega_x \Omega_y$-Norm $> 0 \Rightarrow \Gamma_x \Omega_y$-Norm $> 0$ for $t \in \mathbb{Z}^+$. Thus, the network will still not learn a systematic mapping from the compositional input to compositional output.

### H.3 Proof of Observation 6.2

Finally we prove Observation 6.2: *The fully partitioned network achieves systematicity by learning the lower-rank (compositional) sub-structure in isolation of the rest of the mapping.*

**Proof:** To do this we show that: For all points in the space of datasets with the fully partitioned network: $n_x, n_y, k_x, k_y, r \in \mathbb{Z}^+$ that $\Gamma_x \Omega_y$-Norm $= \Omega_x \Gamma_y$-Norm $= 0 \; \forall \; t \in \mathbb{Z}^+$.

The norm equations for the fully split network are shown in Equations 25 to 26. Clearly from these equations there are no $\Gamma_x \Omega_y$ and $\Omega_x \Gamma_y$-Norms. More specifically, the neural modules learn on restricted datasets based on their connectivity. For example by substituting $k_x = k_y = 0$ for the left module into the full norm equations shown in Equations 17 to 20 the $\Gamma_x \Omega_y$, $\Omega_x \Gamma_y$ and $\Gamma_x \Gamma_y$-Norms in Equations 18, 19 and 20 become 0. Similarly, the right module is restricted to $n_x = n_y = 0$. Thus, the only norm equations which are not 0 will be the $\Omega_x \Omega_y$ -Norm for the left module and the $\Gamma_x \Gamma_y$ -Norm for the right module. The compositional mapping then is not affected by non-compositional components of the data. As a result the fully-partitioned network is systematic by Definition 3.1.

## I Alternative Learning Rules to Gradient Descent

In this work we have relied on gradient descent dynamics since it is the most widely used optimization algorithm for training neural networks. However, a number of alternatives exist and previous work (Cao et al., 2020) has characterized a two-dimensional space of learning rules. The two dimensions correspond to the two parameters $\gamma$ and $\eta$ which determine which optimization algorithm is used. The update equations for the algorithms in the space for a one-hidden layer network are given by Equations 33 and 34, where $||W_{1n}||_2$ depicts the L2 norm is taken along the rows of the $W_1$ matrix, leaving a column vector of dimension equal to the number of hidden neurons.

$$\Delta W_1 = \begin{cases} (1/\tau)W_2^T(\Sigma_{yx} - W_2 W_1 \Sigma_{xx}) + \eta(W_1 X)(X^T - X^T W_1^T W_1) & \text{if } \eta > 0 \\ (1/\tau)W_2^T(\Sigma_{yx} - W_2 W_1 \Sigma_{xx}) + \eta(1/(1 - ||W_{1n}||_2^2))^T(W_1 X X^T) & \text{otherwise} \end{cases} \quad (33)$$

$$\Delta W_2 = (\Sigma_{yx} - W_2 W_1 \Sigma_{xx})W_1^T + \gamma(YY^T - \hat{Y}\hat{Y}^T)W_2 \quad (34)$$

As in (Cao et al., 2020) we focus on four learning rules within this space of learning rules, namely anti-hebbian ($\gamma \to 0, \eta < 0$), contrastive hebbian ($\gamma = 1, \eta = 0$), hebbian ($\gamma \to 0, \eta > 0$) and quasi-predictive coding ($\gamma = -1, \eta = 0$). It is helpful to note that gradient descent is also defined in the space of learning rules with $\gamma = 0$ and $\eta = 0$. Figure 14 depicts the singular value trajectories and the systematic/non-systematic input/output norms for each learning rule compared to gradient descent. From these results we see one primary conclusion, while they cause the network to learn at different speeds, all learning rules result in the network learning along the same modes of variation as with gradient descent. This is evident since all networks achieve a near 0 training error and converge to the same singular values and Frobenius norm values. Thus, they are ultimately learning the same mapping and confined to the dataset statistics. This means that regardless of the learning rule used, from any rule in the whole space defined in (Cao et al., 2020), a dense network will share the $\pi_1$ mode of variation between the compositional and non-compositional components. Thus modularity is required for the same reasons as in Section 6.

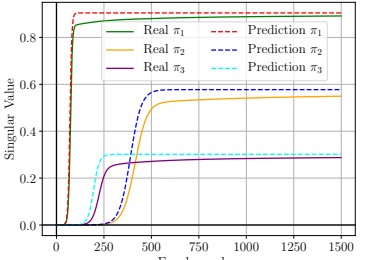
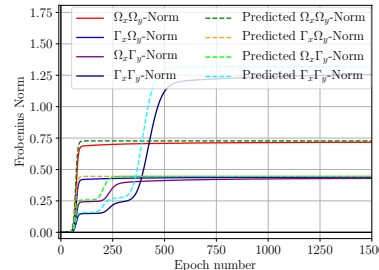

(a) Simulated singular value and norm trajectories for the anti-hebbian (AH) learning rule ($\gamma \to 0, \eta < 0$) compared to the predicted gradient descent (GD) trajectories and norms.

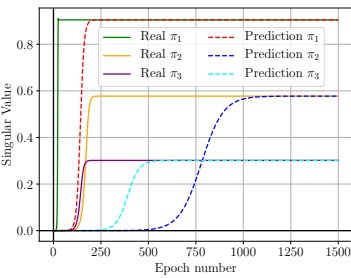
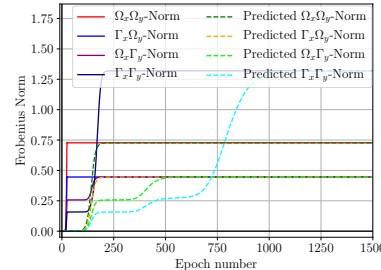

(b) Simulated singular value and norm trajectories for the contrastive hebbian (CH) learning rule ($\gamma = 1, \eta = 0$) compared to the predicted gradient descent (GD) trajectories and norms.

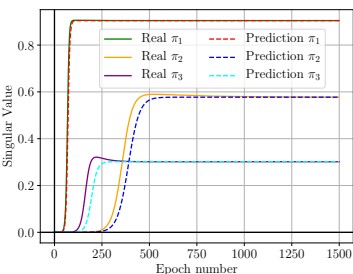
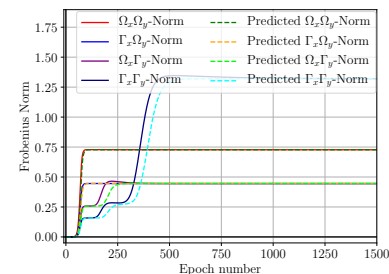

(c) Simulated singular value and norm trajectories for the hebbian (Hebb) learning rule ($\gamma \to 0, \eta > 0$) compared to the predicted gradient descent (GD) trajectories and norms.

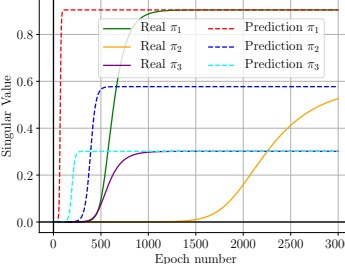
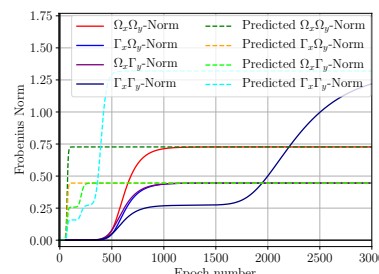

(d) Simulated singular value and norm trajectories for the quasi-predictive coding (PC) learning rule ($\gamma = -1, \eta = 0$) compared to the predicted gradient descent (GD) trajectories and norms.

Figure 14: Simulated singular value trajectories and systematic/non-systematic input/output partitioned norms for the four alternative learning rules compared to the predicted gradient descent trajectories and norms.

## J CMNIST ARCHITECTURE AND HYPER-PARAMETERS

In this section we provide the network architectures for the CMNIST experiments as well as other details of the experimental setup. We scale the non-systematic output labels to help the network learn these labels. For the results of this section a scale of 10 was applied, however, the results are consistent for a wide range of scale values. Increasing or decreasing the scale merely changes the time taken for the same effects to occur. Both the dense and split networks are trained using stochastic gradient descent from small random initial weights sampled from an isotropic normal distribution. No regularization, learning rate decay or momentum is used. We aim to keep the setup as simple as possible while reducing the effects of other implicit or explicit regularizers on the results, since we are comparing the systematic generalization of the networks. The simplicity also aids the comparison between the dense and split network architectures which is the goal of the experiment. Table 3 shows the hyper-parameters used to train both networks, which have the same hyper-parameters, and the two architectures are shown in Figures 15 (dense architecture) and 16 (split architecture).

Table 3: Table showing the hyper-parameters used for the CMNIST experiments.

| Hyper-parameter | Value |
|---|---|
| Step Size | $2e^{-3}$ |
| Batch Size | 16 |
| Initialization Variance | 0.01 |

We describe some further observations in addition to the discussion presented in Section 7 of the main text. Consistent with results in linear networks, when training a dense network there is a portion of the compositional output mapping which is learned at the same time as the non-compositional output mapping (particularly from around epoch 150). In contrast, when using a split architecture the compositional output mapping is learned independently of the non-compositional output mapping and faster, while the non-compositional output mapping struggles to learn. The non-compositional output mapping fails to reach a near-zero error and plateaus around $0.4$ regardless of how long training lasts. Thus, without a compositional output mapping sharing the same hidden layer and helping learning, the non-compositional output mapping is ineffective even at fitting the training data in a reasonable amount of time.

Turning to Figure 4c, we see that for both the dense and split networks the non-compositional output mapping does not generalize to test data. This is to be expected as it is unlikely that the same numbers in the training data would ever appear in the test data. Notably, comparing the compositional output mappings we see that the converged dense network generalizes far worse than the converged split network. It is interesting to note that initially both networks generalize equally well, however, in agreement with our theoretical findings, the dense network is not able to fully learn the compositional output mapping while still maintaining a low generalization error. By learning the non-compositional output mapping the network's hidden layer will become worse for generalization, even for the learned compositional output mapping. In contrast, the split network architecture sees no generalization gap and maintains a near-zero test error from early on in the training. Thus, it is clear that the benefit of using the split network architecture is that it avoids the conflict in its hidden layers from learning to accommodate both the compositional output and non-compositional output mappings.

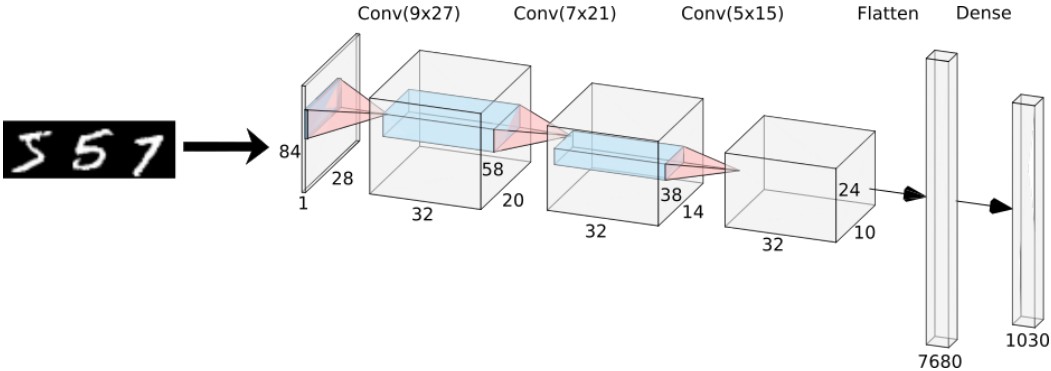

Figure 15: Dense network architecture trained to perform the CMNIST classification task.

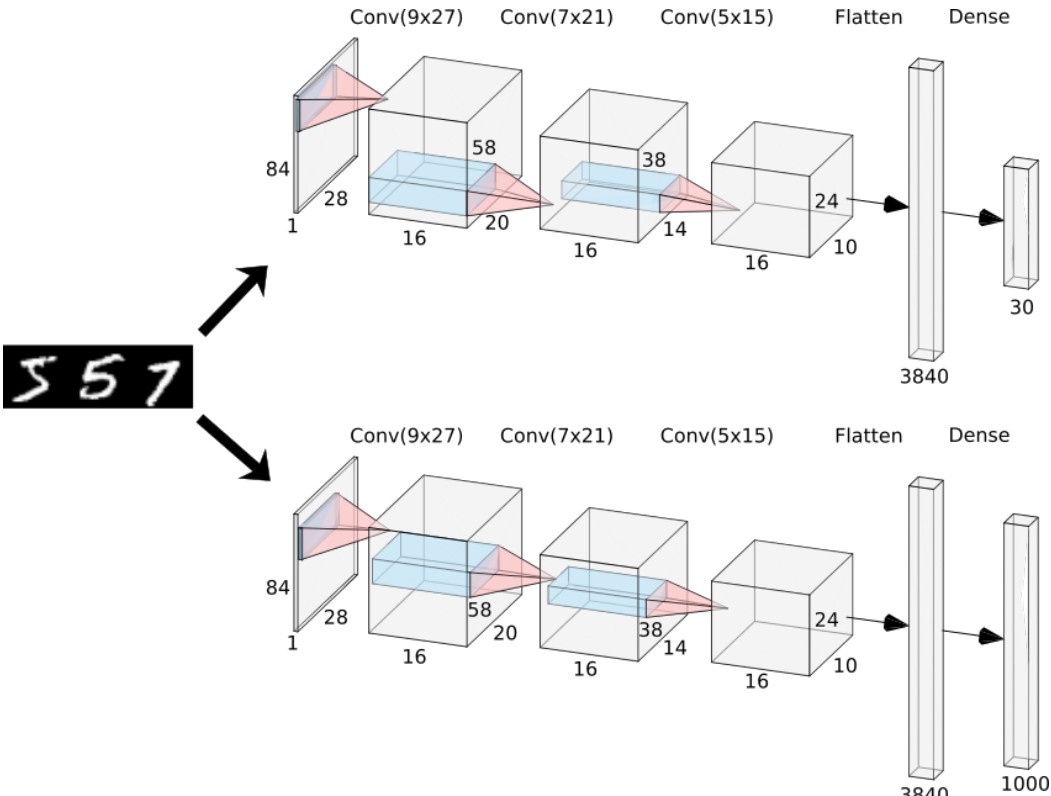

Figure 16: Split network architecture trained to perform the CMNIST classification task.

