# OpenReview forum: "On The Specialization of Neural Modules"
_ICLR.cc/2023/Conference — ICLR 2023 poster_

### Official Review · Reviewer_tTiF · 2022-10-24

**Confidence:** 2
**Correctness:** 4
**Technical Novelty And Significance:** 4
**Empirical Novelty And Significance:** 3
**Recommendation:** 6

**Clarity, Quality, Novelty And Reproducibility:**

- The clarity is the main flaw of the paper, to a significant enough degree that my lack of intuition around the definitions they offer leaves me unable to assess the quality/correctness of the paper properly.
- Very novel to my knowledge.
- The lack of clarity might pose a challenge for reproduction of the results.

**Strength And Weaknesses:**

Strengths:
- The idea here is really exciting to me. I would love to see more of this kind of exact analytic result around compositional behavior. I remember originally reading Saxe 2019 and wondering how it would interact with compositional or systematic behavior, and this approach seems like a really solid attempt at that.
- A number of steps taken to formalize modularity and systematicity, and the techniques for analysis that result, are innovative and likely to be useful in future work. In particular, I appreciate the use of the compositional properties of the dataset as a parameter when calculating gradient trajectories.
- The authors do test whether the predictions theoretically made in the linear setting also apply in a nonlinear setting.
- “For example, a module with no hidden layer connections to the non-compositional output components is learning on a dataset with ky = 0.” This is such a clean way of formalizing modularity!
- Again, I want to restate the serious potential of the formalisms that they offer and the techniques that they are using in their analysis, even if I think their presentation definitely needs work and the definitions might need some refinement.

Weaknesses:
- The theoretical results are only in the linear setting. I think that they might still be really valuable as a starting point for understanding more realistic networks, but it’s certainly a limitation in this work, and it’s not obvious to me how future work might adapt these methods, given their reliance on the linearity of module interaction. Although there are empirical results in a nonlinear setting that, given that it is only MNIST, these interactions are likely to be more linear than they would be in a network that trained for a longer period of time, so they might align better with the predictions made in the theoretical setting than a modern network would.
- “Neither implicit biases in learning dynamics, nor all but the most stringent modularity, caused networks to exploit compositional sub-structure in the data.” — again I’m not sure whether this is going to apply outside of these relatively low resource, somewhat linear, empirical settings.
- I often struggled to follow the intuitions behind the definitions. Examples of dataset that are combinatorial only in the input or only in the output, and ones that are combinatorial in both input and output, would have helped a lot with this. Could you give me some examples now? I’m really still stuck on the intuitions behind some of these definitions.
- I’m not even entirely clear on the intuition of the difference between the combinatorial and non combinatorial features, that could also really do with an example.
- Likewise, it was hard for me to grasp the intuitions yielded by the trajectory results (eq 5-9). What are the implications in terms of data efficiency? Let’s assume that most readers are not particularly effective at simulating and comparing function curves in their heads. All we have to go on is Fig 2, and it’s clear something is going on there but it’s a lot harder to understand what it is that you want me to take away.
- In general, I would really like to see examples of intuitions. Using the inspiration of Saxe 2019 as an example, there are running examples of specific hierarchical semantic classes throughout the text. This work would be greatly supported by an equivalent running example of compositional data of varying sorts.
- In definition 3.1, it’s specified that systematic generalization is only present if it “facilitates generalization from fewer training examples”. However, the results give in don’t seem to have anything to do with data scarcity, so I’m not sure how they link into this part of the definition. Could you spell out the implications in particular of these results in figure 2 with respect to the data efficiency alluded to in definition 3.1?
- I think that you really need to be testing generalization behavior or the amount of data required. In particular, you should be looking at splits that constrain the combinations exposed during training, as is typical in systematicity research.

Questions and minor issues:
- Are the rules for each of these datasets randomly generated?
- Without a more solid intuition about your definitions, it’s hard for me to say whether 3.1 would apply to behavior I wouldn’t think of as systematic personally. For example, if the matrix is a sum of the (non combinatorial?) “true” data and a noise matrix formed the covariance matrix, would that end up fitting into this definition? It doesn’t seem that effectively representing the underlying true data is a systematic behavior, but it seems like it might fit into this definition.
- “it is possible to have a near infinite number of resultant combinations” — really?
- Figure 3: There might be some notational inconsistency here. Previously, only the asymptotic bound was defined with $\pi^{ss}$, and I don’t think I’ve seen this particular variable name show up already.
- “To do this we calculate the Frobenius norm of the network’s mapping between different subsets of input and output components.” This needs a much more clear explanation of what form the mapping takes. What are you defining as a mapping?
- I really don’t like how you index figure references! Every time I see “Figure 3b)” I cringe at the unmatched parenthesis.

**Summary Of The Paper:**

This paper works on expanding the exact training dynamics analysis of Saxe et al 2019, which considers interrelated semantic groupings, to also cover systematic behavior. For this analysis, they consider the effect of combinatorial behavior in the dataset on the resulting gradient dynamics. In particular, they consider how a deep linear model without internal modular structure compares to one with modular structure in its trajectories. The result is an attempt to precisely characterize the importance of implicitly modular network structure in systematic generalization.


**Summary Of The Review:**

Because of the lack of clarity around the intuitions underlying the math, I don’t feel confident assessing the correctness of the paper. This is reflected in my confidence score, although I find the concepts and the general approach of the paper very exciting so my score reflects both the promising concept and the need for grounding and intuition.

**After discussion:**

I think I agree with reviewer 83KP that this would be better served by a journal where the authors could fully explain their work within the main body of the paper. I have raised my score from 5 to 6 in response to the authors' attempts to clarify the writing, but I just think that there is a ceiling on how good this paper can get with only nine pages. The work here is impressive, and I would rather see it published in a form that serves it well.

---

> ### Author Response · Authors · 2022-11-14
> **Response to Official Review of Paper5201 by Reviewer tTiF (Part 1 of 5)**
>
> We thank the reviewer for their helpful comments and kind words. We find the reviewer's view of the potential impact very motivating and hope that the discussion during the rebuttal period will improve the paper. We respond to each comment made in the review individually below (some related points have been grouped together). We have revised our paper based on the reviewer's feedback and have indicated these changes in red to make further discussion easier.
>
> **Comment**: The theoretical results are only in the linear setting. I think that they might still be really valuable as a starting point for understanding more realistic networks, but it’s certainly a limitation in this work, and it’s not obvious to me how future work might adapt these methods, given their reliance on the linearity of module interaction. Although there are empirical results in a nonlinear setting that, given that it is only MNIST, these interactions are likely to be more linear than they would be in a network that trained for a longer period of time, so they might align better with the predictions made in the theoretical setting than a modern network would.\
> **Comment**: “Neither implicit biases in learning dynamics, nor all but the most stringent modularity, caused networks to exploit compositional sub-structure in the data.” — again I’m not sure whether this is going to apply outside of these relatively low resource, somewhat linear, empirical settings.\
> **Response**: The linearity of our analysis is indeed a limitation. However, few theoretical tools currently exist which are able to incorporate non-linearity *and* feature learning. Feature learning dynamics are essential because only through feature learning can neural modules specialise. Thus, this appears to be one of very few theoretical frameworks presently available for studying the specialisation of neural modules. We hope our results will serve as an important prerequisite to future analyses of more complicated settings, perhaps using the recent Neural Race Reduction formalism that builds off of deep linear networks [1]. Notably, in this work we are investigating an effect which has already been observed on far more complex tasks/architectures/datasets. Specifically Neural Module Networks [2], Recurrent Independent Mechanisms [3] and Compositional Recursive Learners [4] have all displayed cases where modules fail to specialise on complex datasets such as SCAN [5], gSCAN [6] and CLEVR [7]. Through our experiments with CMNIST we display that convolution, non-linearity and a more naturalistic setting are not enough to guarantee module specialisation. We hope our results will ultimately contribute to understanding the aspects of more complex models that might yield systematicity, such as perhaps seen in large language models. Ultimately our results display the difficulty which arises due to spurious correlations in the input or output space of a dataset. Finally, we now include a discussion in the conclusion (in green following a suggestion from Reviewer rTdC) on how Definition 3.1 and our analysis can be interpreted for more complex datasets.\
> $[1]$ Saxe, Andrew, Shagun Sodhani, and Sam Jay Lewallen. "The neural race reduction: Dynamics of abstraction in gated networks." International Conference on Machine Learning. PMLR, 2022.\
> $[2]$ Andreas, Jacob, et al. "Neural module networks." Proceedings of the IEEE conference on computer vision and pattern recognition. 2016.\
> $[3]$ Goyal, Anirudh, et al. "Recurrent independent mechanisms." arXiv preprint arXiv:1909.10893 (2019).\
> $[4]$ Chang, Michael B., et al. "Automatically composing representation transformations as a means for generalization." arXiv preprint arXiv:1807.04640 (2018).\
> $[5]$ Lake, Brenden, and Marco Baroni. "Generalization without systematicity: On the compositional skills of sequence-to-sequence recurrent networks." International conference on machine learning. PMLR, 2018. \
> $[6]$ Ruis, Laura, et al. "A benchmark for systematic generalization in grounded language understanding." Advances in neural information processing systems 33 (2020): 19861-19872.\
> $[7]$ Johnson, Justin, et al. "Clevr: A diagnostic dataset for compositional language and elementary visual reasoning." Proceedings of the IEEE conference on computer vision and pattern recognition. 2017.

---

> > ### Author Response · Authors · 2022-11-14
> > **Response to Official Review of Paper5201 by Reviewer tTiF (Part 2 of 5)**
> >
> > **Comment**: I often struggled to follow the intuitions behind the definitions. Examples of dataset that are combinatorial only in the input or only in the output, and ones that are combinatorial in both input and output, would have helped a lot with this. Could you give me some examples now? I’m really still stuck on the intuitions behind some of these definitions.\
> > **Comment**: I’m not even entirely clear on the intuition of the difference between the combinatorial and non combinatorial features, that could also really do with an example.\
> > **Comment**: In general, I would really like to see examples of intuitions. Using the inspiration of Saxe 2019 as an example, there are running examples of specific hierarchical semantic classes throughout the text. This work would be greatly supported by an equivalent running example of compositional data of varying sorts.\
> > **Response**: We thank the reviewer for their assistance with improving the clarity of our work. We have taken a number of steps to address the above concerns. Firstly, we have improved Figure 1(a) to offer a better conceptual example of compositionality which hopefully fixes intuitions with the mouse-maze navigation task example. Second, we have improved upon Figure 1(b) by adding many more annotations as well as a key which summarises the notation used in the work. Finally, we have added a new section in Appendix A with a new figure (Figure 5) that displays five example datasets and their various locations in our space of datasets, which we hope now give examples of datasets as requested by the reviewer. Appendix A as a whole provides a motivating example for our space of datasets and definition of systematicity. We hope this section will greatly improve clarity and intuition on these concepts and welcome further suggestions.
> >
> > **Comment**: Likewise, it was hard for me to grasp the intuitions yielded by the trajectory results (eq 5-9). What are the implications in terms of data efficiency? Let’s assume that most readers are not particularly effective at simulating and comparing function curves in their heads. All we have to go on is Fig 2, and it’s clear something is going on there but it’s a lot harder to understand what it is that you want me to take away.\
> > **Response**: Thank you, following a suggestion from Reviewer 83KP we have now added proof sketches which make it clear how the Observations result from our equations--these summarise the essential take-aways. We hope this will greatly improve their clarity. We present Equations (5)-(9) to show only that the five dataset parameters are present in the SV equations, and that we have explicit formulae for them--the resulting behavior is intricate and is unpacked in Fig 2 and subsequent Observations. This is important as formalising the dataset in a manner where the effect on training dynamics can be determined is a contribution of this work. Figure 2 serves to demonstrate the accuracy of these equations and has the main take-away point summarised in the caption: ``Deep networks show distinct stages of improvement over learning. However, at no point is a systematic mapping learned''. We have now moved this statement to the beginning of the caption to make sure it is clearly the primary point of this figure. That is, we have made it clear that the main comparison for readers is between our theoretical predictions and the simulations (dotted lines compared to the solids lines) rather than comparing trajectories themselves--this is a visualisation of the correctness of our theoretical results (for these parameter settings, at least).

---

> > > ### Author Response · Authors · 2022-11-14
> > > **Response to Official Review of Paper5201 by Reviewer tTiF (Part 3 of 5)**
> > >
> > > **Comment**: In definition 3.1, it’s specified that systematic generalization is only present if it “facilitates generalization from fewer training examples”. However, the results give in don’t seem to have anything to do with data scarcity, so I’m not sure how they link into this part of the definition. Could you spell out the implications in particular of these results in figure 2 with respect to the data efficiency alluded to in definition 3.1?\
> > > **Comment**: I think that you really need to be testing generalization behavior or the amount of data required. In particular, you should be looking at splits that constrain the combinations exposed during training, as is typical in systematicity research.\
> > > **Response**: We have added the empirical results from the second comment into a new Appendix A. From this motivating example it should now be clearer how beneficial systematic generalisation is and how our definitions align with which networks generalise to test data when a train-test data split is used. Additionally, data sparsity is the topic of Section 3.1. Specifically, a network only needs to see as many training examples from a dataset as the rank of the dataset's covariance matrix. Thus, the lower-rank the covariance matrix the less training examples are required to generalise to the entire dataset (the more data sparsity can be tolerated). Thus, the notions of data sparsity and generalisability come directly from our discussion of rank. We have also added this point to Appendix A [how systematicity promotes generalisation] and direct the reader to Appendix A in Section 3.1 (we say ``see Appendix A for a motivating example of this point and for more intuition on Definition 3.1''). As far as how a train-test data split affects the dynamics, this is an interesting direction for future work--we show here that even with large amounts of data, systematicity is not learned in our setting. When a training set is sampled, the dynamics will necessarily be different regardless of the architecture used. Essentially, subsampling such that modules specialise is the method employed by Neural Module Networks (a module is only used for data points with the colour red for example). The contribution of our work is to show that regardless of data quantity, if this subsampling (module allocation) is not perfect then the module will not specialise. Please see the green text in the Discussion (Section $8$) for more on how our analysis relates to complex architectures.
> > >
> > > **Comment**: Are the rules for each of these datasets randomly generated?\
> > > **Response**: No, the rule (mapping) depends on the covariance matrices $\Sigma^x$ and $\Sigma^{yx}$ which are influenced indirectly through the five dataset parameters. This is the point of formalising the space of datasets: we can express the rule of a network (and how it learns the rule) in terms of the dataset parameters (how compositional vs non-compositional the dataset is) for any dataset in the space.

---

> > > > ### Author Response · Authors · 2022-11-14
> > > > **Response to Official Review of Paper5201 by Reviewer tTiF (Part 4 of 5)**
> > > >
> > > > **Comment**: Without a more solid intuition about your definitions, it’s hard for me to say whether 3.1 would apply to behavior I wouldn’t think of as systematic personally. For example, if the matrix is a sum of the (non combinatorial?) “true” data and a noise matrix formed the covariance matrix, would that end up fitting into this definition? It doesn’t seem that effectively representing the underlying true data is a systematic behavior, but it seems like it might fit into this definition.\
> > > > **Response**: Please see revised Appendix A for further intuition on Definition 3.1. Based on the reviewer's comments we now include several examples in light of our definition in this appendix, and believe that it helps build intuition. Extending our results here to incorporate noise is an important direction of future work. In this example ignoring the noise would not correspond to a systematic mapping as we would end up back at the setup presented in this work (we show that systematicity does not emerge naturally in these cases). Thus, the natural implication of our work is that noise tolerance is not enough to achieve systematicity (at least not in the space of datasets with a linear network). Thus, the reviewer's intuition is correct and our definition does not imply that robustness to noise is sufficient. The example in revised Appendix A should also make clear that, by Definition 3.1, systematic generalisation is a more sophisticated form of generalisation which relies on networks identifying sub-structure to generalise. To provide a potentially helpful second example: suppose the matrix is the compositional input and output data (thus being lower rank than the number of examples) with a noisy set of features appended (thus increasing the rank of the problem). Ignoring the noisy features in this example and just relying on sub-structure would be an example of systematicity. We note that, while this definition captures some settings and aspects of systematicity (particularly in linear settings), we expect different definitions to be necessary to address the full scope of the topic. We emphasise that the difference between generalisation and systematic generalisation is a point which appears naturally in our motivating example in the revised Appendix A and thank the reviewer again for leading us to add this section. We believe it will be helpful to a number of readers.
> > > >
> > > > **Comment**: “it is possible to have a near infinite number of resultant combinations” — really?\
> > > > **Response**: This comment was discussing how generalisability grows in accordance with the number of compositional features. Put simply, if a systematic mapping were to be trained on a dataset with $n_x$ compositional features it would only need to see $n_x$ datapoints to generalise to all $2^{n_x}$ datapoints. So for $n_x = 3$ it generalises to $2^3 = 8$ datapoints. If $n_x = 20$ the network generalises to $2^{20} = 1048576$ combinations (which we termed as near infinite in the hypothetical case of increasing $n_x$). We have changed this potentially misleading comment and now say that generalisability grows exponentially. Additionally, please see our above comments for a discussion of rank and how many training examples are needed for generalisation.
> > > >
> > > > **Comment**: Figure 3: There might be some notational inconsistency here. Previously, only the asymptotic bound was defined with $\pi^{ss}$, and I don’t think I’ve seen this particular variable name show up already.\
> > > > **Response**: We thank the reviewer for pointing this out. We have changed the notation in Section 4 slightly (also as a result of some discussion with Reviewer 83KP) and now it is clear that $\pi_i^*$ is an asymptote of the SV trajectory $\pi_i$. Specifically we say ``three asymptotes $\pi_1^*$,$\pi_2^*$ and $\pi_3^*$ (the final point for each SV trajectory $\pi_1$,$\pi_2$ and $\pi_3$)'' We hope this point is clearer now.
> > > >
> > > > **Comment**: “To do this we calculate the Frobenius norm of the network’s mapping between different subsets of input and output components.” This needs a much more clear explanation of what form the mapping takes. What are you defining as a mapping?\
> > > > **Response**: A mapping is the learned function by the network from a set of input components to a set of output components. Thank you for point out where this is lost, we have clarified this in the revision. Specifically we now say: ``To do this we calculate the Frobenius norm of the network’s mapping (the function implemented by the network transforming inputs to outputs)'' Finally, we ask the reviewer to please see the addition in the conclusion (in green following a suggestion from Reviewer rTdC) on how the input/output partitioning may extend to more complex datasets.
> > > >
> > > > **Comment**: I really don’t like how you index figure references! Every time I see “Figure 3b)” I cringe at the unmatched parenthesis.\
> > > > **Response**: Many thanks, we have changed this.

---

> > > > > ### Author Response · Authors · 2022-11-14
> > > > > **Response to Official Review of Paper5201 by Reviewer tTiF (Part 5 of 5)**
> > > > >
> > > > > We thank the reviewer once again for the assistance in improving the clarity of our work. We believe the resulting changes to Figures 1 and 2 and the addition of Appendix A in particular greatly improve the conceptual clarity of our work. We look forward to engaging further and if there are any more changes we can make to improve the paper we are keen to make them.

---

> > ### Comment · Reviewer_tTiF · 2022-11-15
> > **linearity**
> >
> > I agree that this weakness is one shared by most theoretical work on training dynamics! The fact that you empirically test your claims outside of a linear setting already reaches beyond the typical work of this sort. I only mentioned it in my review to gesture at this common weakness; I would not reduce your score on these grounds. I only reduced your score in response to the clarity of the writing, and I feel you've already improved readability with your current modifications. It might be that this work is still more suitable for a journal where you have more space to explain your formalisms thoroughly and clearly, but I will wait to reassess my score after your final revision.

---

> ### Comment · Reviewer_tTiF · 2022-11-16
> **raising my score**
>
> I want to emphasize that this is fantastic work, and I'm really excited to see it published. But I don't actually think a conference is an appropriate venue. Even after revision, for example, the new figure 1 that is supposed to present all of the intuitions is uninterpretable from just the caption and paper. Instead, I felt like I had to reconstruct your thinking from the bottom up in order to understand the formalism. I do appreciate your new appendix A, though. I just think that a brief version of something like appendix A would ideally fit into the mean body of the paper, so that the reader can follow the fundamental concepts without having to read the appendix; I don't feel that the main body is self contained.
>
> I think that you should probably submit this work to TMLR or JMLR, where you would have adequate space to explain everything. I suggest using Saxe et al. to guide you through the degree to which you should explain each concept; that paper was in PNAS, and longer than 9 pages even with a more compressed style template. (And their hierarchical semantics is generally easier to explain than your notions of systematicity.) You've put effort into improving readability, but I think it will be a better paper without the constraints of conference length. I'm raising my score to a 6 to reflect your improvements, and I definitely would be happy to see it published at ICLR---I think people should read it! But I believe that having an extra page or two could turn this into a great paper, by presenting clear intuitions, laying the groundwork of how to think about systematicity formally, and walking through your definitions.

---

> > ### Author Response · Authors · 2022-11-17
> > **Many thanks, we will continue to improve on clarity.**
> >
> > Many thanks for considering our responses and confirming that our revisions have been a step in the right direction. We agree that Appendix A is a particularly helpful addition and we will continue to work on clarity up to the final revision deadline. Finally, we thank the reviewer for their kind comments in support of the overall work. It is both motivating and gratifying, and we hope that a broad group of readers will find this work helpful.

---

### Official Review · Reviewer_83KP · 2022-10-24

**Confidence:** 3
**Correctness:** 3
**Technical Novelty And Significance:** 3
**Empirical Novelty And Significance:** 2
**Recommendation:** 5

**Clarity, Quality, Novelty And Reproducibility:**

Clarity: The paper is too hard to read, as it tries to compress too much information in the body while skipping over most of the actual analysis (that is in the appendices). The setup is also needlessly complex, and could be simplified greatly while keeping the core message intact.

Quality: As far as I can tell, the proofs appear to be correct, though I only skimmed some of them. However, many results are given without any proof and we only get to see the final answer, e.g., the derivation of the singular value decomposition equations in App. C, or the partitions of the forbinuous norm in App. D. The authors should provide more details on how they got to these results (though I did verify parts of it on my own for simple cases, so it seems to be correct).

Novelty: There is novelty in the simple setup that allows us to pin-point the issue of decomposability vs end-to-end learning. However, it is not completely novel as there are similar approaches in prior works, though the analysis here appears to be a bit more general.

Reproducibility: As mentioned, many details are missing from the technical proofs that would make it difficult to verify. As for the experiments, they were straightforward and simple.

**Strength And Weaknesses:**

Strengths:

* The topic is of significant importance to the community. Many works have tackle this curcial topic of decomposable vs "end to end" learning, or systematic generalization as this paper calls it, and it is still an area where our understanding is lacking. Prior works have used more restrictive analysis, which this work (based on the recent development of gradient flow analysis) alleviates.

* The paper contains many intermediate results that could be of independent interest for further investigating this topic.

Weaknesses:

* The paper is too dense and might be better served as a journal paper without a strict page limit. Most of the theoretical analysis is demoted to the appendices, and just reading the body of the paper can be very confusing. Some sections end without any point or context, for example, the training dynamics solely discuss the dynamics of the singular values, but they should've mentioned first why we care about the singular values (discussed in the next section), and how the asymptotic solution converges to the correct function for predicting Y given X (It's possible to show this with some basic linear algebra, but it's never mentioned). Alternatively, the authors should consider removing and simplifying its various sections, while providing proof sketches for the analysis. Even the problem setup is hard to understand and there are many things that are left poorly described or explained, e.g., what is the difference between the various non-combinatorial features (aren't they all the same identity matrices?)? How are the subset of the combinatorial features sampled, and how does this distribution affect the results? Does the number of non-combinatorial output features depend on the number of combinatorial output features?

* The basic setup of the paper seems needlessly intricate. As I see it, this is simply the problem of learning the identity function, where we are given "side information" on the *preferred* binary representation of the input, and you examine how these input-outputs are represented and the relation to the architecture. The extra copies of the non-combinatorial features, or the subset selection seem like complications that don't really add anything to the discussion. Moreover, I think that there is some subtlety that the authors need to discuss more carefully. Namely, in their theoretical setup they merely consider learning the identity (though it is not presented as such), but simple random linear networks would "learn" such mapping over the "challenging" one-hot case immediately -- specifically, if W_1 is normally distributed with 1/sqrt(d) standard deviation, where d is the input dimension, and W_2 is constrained s.t. W_2 = W_1^T, then W_1 W_2 is approximately the identity even with the number of hidden units is as low as log(d). In other words, a randomly initialized network (not that far from the conventional initialization scheme) could approximately solve this case.

* The paper does not cite or discuss highly related prior works. As mentioned above, there is not much difference between the topic of "systematic generalization" / compositionality to the slightly "older" (in ML terms) topic of "decomposability vs end-to-end learning" that has been studied by others (e.g., [1, 2, 3]). In fact, a very similar setup and experiment has been proposed in [2], and analysed both theoretically and empirically, albeit using a different analysis and a more limiting set of assumptions. A thorough discussion of these works is missing.

[1] - Shalev-Shwartz et al., On the Sample Complexity of End-to-end Training vs. Semantic Abstraction Training, arXiv preprint arXiv:1604.06915, 2016.

[2] - Shalev-Shwartz et al., Failures of Gradient-Based Deep Learning, ICML 2017.

[3] - Wies et al., Sub-Task Decomposition Enables Learning in Sequence to Sequence Tasks, arXiv preprint arXiv:2204.02892, 2022. (Given it's a relative new preprint I don't expect it to be cited, but nevertheless I place it here for context).

**Summary Of The Paper:**

The paper proposes a theoretical framework for studying systematic generalization that considers the relationship between the structure in the dataset and the neural network architecture used to learn its input-output mapping. It defines two kinds of input/output features, either combinatorial features that are binary values {-1, 1} values, or non-combinatorial features that map all combinatorial features to one large 2^n one-hot vector according to the binary string. The output corresponding to the input is defined as a random subset of the combinatorial features and the non-combinatorial features. In essence the task is to learn the identity function where the subset of features is unknown apriori, where both input and output space contain redundant representations of the same information (the one-hot vector exactly identifies the combinatorial features in the input).

Given this description of the input-output X, Y matrices, they define systematic generalization as leveraging a low-rank substructure of the correlation matrix YX^t. They then proceed to provide the optimization dynamics for deep linear networks using gradient flow type of analysis, and show how the dynamics depend on the singular values of the correlation matrices YX^t and XX^t, for which they provide an exact closed form description relating the singular values to the number of feature types the dataset is constructed of. Using these training dynamics, they proceed showing how the correlation between different input-output splits changes during training (combinatorial to combinatorial feature, combinatorial to non-combinatorial, etc.), and come to the conclusion that dense networks cannot learn a systematic mapping between combinatorial features (i.e., that does not depend on the non-combinatorial inputs). However, when defining a simplified form of a neural module network, where the network completely separates the handling of the two kinds of features, then the network is able to learn the systematic mapping. Finally, the authors propose a simple empirical demonstration of these ideas in the form of predicting 3 MNIST digits, where the authors consider two cases, where the goal is to predict the 1000 combinations as 1000 classes or as 3 separate 10 class problems. The authors test this setup both for the dense case as well as for the split-network case where networks process each digit separately. The experiments show a clear benefit to learning problems when they are presented in a decomposable fashion, both for the input-output representation and for the chosen architecture.

**Summary Of The Review:**

In its current state, the submission is a bit below the threshold for acceptance. It has severe clarity issues that obscure its theoretical contributions, though I believe these could be corrected with a revision.

---

> ### Author Response · Authors · 2022-11-14
> **Response to Official Review of Paper5201 by Reviewer 83KP (Part 1 of 4)**
>
> We thank the reviewer for their helpful comments and for seeing the potential significance of this work. We find this very motivating and hope that the discussion during the rebuttal period will address the concerns. We respond to each comment made in the review individually below. Additionally, we have revised our manuscript based on the reviewer's feedback and have indicated these changes in blue to make further discussion easier.
>
> **Comment**: The paper is too dense and might be better served as a journal paper without a strict page limit. Most of the theoretical analysis is demoted to the appendices.\
> **Response**: As noted in the response to all reviewers, we have taken steps to improve the technical and conceptual clarity in the main text. For example every Observation now has a corresponding proof outline, as the reviewer suggested. In doing so we have restructured some sections, and also made sure sections have a clear point and lead better into the next. We hope this has greatly improved the readability and clarity of the main text. In addition, we believe the changes in structure make each section less dense, while still covering the same content. Finally, we have amended figures to make our setting more intuitive. In our view, the issue of clarity is paramount even for a journal paper, which often requires a similar format and page limit. We hope the changes we have made will make the work accessible for a general ML audience who we believe will benefit from our work. We thank the reviewer for assisting us in enhancing the clarity of our work and hope to further improve this during the rebuttal period.
>
> **Comment**: Some sections end without any point or context, for example, the training dynamics solely discuss the dynamics of the singular values, but they should've mentioned first why we care about the singular values (discussed in the next section), and how the asymptotic solution converges to the correct function for predicting Y given X (It's possible to show this with some basic linear algebra, but it's never mentioned).\
> **Response**: Thank you for the comment, we have now made sure that every section ends with a conclusion which also links it to the following section. For example in the linear dynamics section we conclude by saying: "In the following sections we utilise these equations for the training dynamics to analyse to what extent neural networks naturally specialise and display systematicity''. Additionally, we have amended the explanation of the dynamics which emphasise the role of the singular values in learning the covariance between the input and output data. Specifically we now state that ``This trajectory corresponds to the network learning the covariance between input and output data - the correct mapping''. This also now motivates the use of the SVD (Equations (4) to (9)) which follows in the next paragraph.
>
> **Comment**: Alternatively, the authors should consider removing and simplifying its various sections, while providing proof sketches for the analysis.\
> **Response**: We once again thank the reviewer for their assistance with the clarity of this work. We have now added proof sketches for every Observation which provides the intuitive reason for why each observation arises from our equations (specifically the norm equations: $\Gamma_x\Omega_y$-Norm and $\Omega_x\Gamma_y$-Norm). Additionally we have linked each observation back to Figure $3$ more clearly in cases where the link was not clear in the original version. We also hope that the clear conclusions and links we have added to each section assist in making the logic of our section breaks clear.

---

> > ### Author Response · Authors · 2022-11-14
> > **Response to Official Review of Paper5201 by Reviewer 83KP (Part 2 of 4)**
> >
> > **Comment**: Even the problem setup is hard to understand and there are many things that are left poorly described or explained, e.g., what is the difference between the various non-combinatorial features (aren't they all the same identity matrices?)? How are the subset of the combinatorial features sampled, and how does this distribution affect the results? Does the number of non-combinatorial output features depend on the number of combinatorial output features?\
> > **Response**: We thank the reviewer for raising this essential point. We have made the various points raised more clear in Section $3$ and thank the reviewer for being specific in their suggestions. Thus, we now state that the compositional outputs are sampled uniformly from the compositional inputs but the results will work for any sampling. We've also made it explicit that the number of non-compositional outputs in each block depends on $n_x$ (the number of compositional input features) since this determines how many data points are in the dataset. Since each data point has one corresponding non-compositional feature in a non-compositional block the number of non-compositional features matches the number of data points. Thus we have shown in the text that each non-compositional block has shape: $I_i \in \\{0,1\\}^{2^{n_x} \times 2^{n_x}}$. We also think that making this explicit shows that each non-compositional block is identical. Finally, we have also made improvements to Figure $1$ which we believe now demonstrates how each dataset parameter affects the dataset. The new key in Figure $1$ should also help readers to have a single simple point to return to find our notation and understand what each symbol corresponds to. Additionally, the new Appendix A provides extended intuition on the space of datasets and the link to systematicity.
> >
> > **Comment**: The basic setup of the paper seems needlessly intricate. As I see it, this is simply the problem of learning the identity function, where we are given "side information" on the preferred binary representation of the input, and you examine how these input-outputs are represented and the relation to the architecture. The extra copies of the non-combinatorial features, or the subset selection seem like complications that don't really add anything to the discussion.\
> > **Response**: We believe there may be a confusion here, the network does not learn the identity mapping. If a network were to learn the identity mapping then this would mean the network ignores non-compositional input components to determine the compositional outputs. That is, it would be systematic. That this doesn't happen is of general interest to the field of systematic generalisation and ML in general. Instead, the network learns links from the non-compositional inputs to the compositional output, and therefore deviates from an identity mapping. We intentionally create a setting in which there are a spectrum of redundant solutions (one of which is the identity mapping), and examine which solutions the dynamics in fact converge to. Regarding the setup being complex, we agree that our main point would be conveyed by datasets that do not have extra copies or subset selection. However, we include these more complex options for two reasons. First, we use them to show that our results are robust within a wider (though still limited) class of datasets. We note that all five of the dataset parameters appear in the singular value and norm equations. Thus, all parameters have a tangible effect on the network dynamics. Second, we use them to analyse dynamics in alternative architectures, which often yield sub-problems corresponding to these more complex dataset parameters. For instance, an alternative to having extra copies of the non-combinatorial features is to take $k_x,k_y \in \\{0,1\\}$ (that is, be indicator variables depicting the absence or presence of non-compositional features). This restriction to 0 or 1 copies, however, would not change any of the equations, discussion or conclusions; and the $k_x$ and $k_y$ parameters are useful for analysing modularity formally. As Reviewer tTiF notes: ` “For example, a module with no hidden layer connections to the non-compositional output components is learning on a dataset with ky = 0.” This is such a clean way of formalizing modularity! '. Finally, the subset selection section is included in the appendix as it shows another clear example of how modularity can be viewed as imposing a limitation on the space of datasets (similar to the point Reviewer tTiF notes above) and also that only perfect modularity achieves systematicity. We hope our clarity revisions bring these points forward.

---

> > > ### Author Response · Authors · 2022-11-14
> > > **Response to Official Review of Paper5201 by Reviewer 83KP (Part 3 of 4)**
> > >
> > > **Comment**: Moreover, I think that there is some subtlety that the authors need to discuss more carefully. Namely, in their theoretical setup they merely consider learning the identity (though it is not presented as such), but simple random linear networks would "learn" such mapping over the "challenging" one-hot case immediately -- specifically, if $W_1$ is normally distributed with 1/sqrt(d) standard deviation, where d is the input dimension, and $W_2$ is constrained s.t. $W_2$ = $W_1^T$, then $W_1 W_2$ is approximately the identity even with the number of hidden units is as low as log(d). In other words, a randomly initialized network (not that far from the conventional initialization scheme) could approximately solve this case.\
> > > **Response**: We agree this point is subtle: with appropriate initialisation, the network could `solve' our task without learning. We are specifically interested in the feature learning regime, which behaves differently from these large initialisations (see for instance $[1-3]$). However, more fundamentally, we have chosen our datasets to be simple and interpretable, but our analysis applies to a more general situation for which a random initialisation fails. In particular, consider the broader space of datasets in which compositional inputs, compositional outputs, non-compositional inputs, and non-compositional outputs in our space of datasets are each rotated by individual orthogonal mappings. As one specific example, consider the dataset formed by applying a permutation to the compositional outputs. This change affects only the singular vectors of the task (also permuting them relative to the original task), not the singular values. This dataset can no longer be solved by learning an identity transformation, as the network must learn to implement the required permutation--but it will do so with identical learning dynamics and systematicity properties. A random initialisation cannot know this desired permutation, which can only be learned through error feedback. We now include this discussion in Appendix C and call to it where we introduce the linear network dynamics (Section $4$).\
> > > $[1]$ Geiger, Mario, et al. ``Disentangling feature and lazy training in deep neural networks." Journal of Statistical Mechanics: Theory and Experiment 2020.11 (2020): 113301.\
> > > $[2]$ Jacot, Arthur, Franck Gabriel, and Clément Hongler. "Neural tangent kernel: Convergence and generalization in neural networks." Advances in neural information processing systems 31 (2018). \
> > > $[3]$ Sirignano, Justin, and Konstantinos Spiliopoulos. "Mean field analysis of neural networks: A law of large numbers." SIAM Journal on Applied Mathematics 80.2 (2020): 725-752.
> > >
> > > **Comment**: The paper does not cite or discuss highly related prior works. As mentioned above, there is not much difference between the topic of "systematic generalization" / compositionality to the slightly "older" (in ML terms) topic of "decomposability vs end-to-end learning" that has been studied by others (e.g., [1, 2, 3]). In fact, a very similar setup and experiment has been proposed in [2], and analysed both theoretically and empirically, albeit using a different analysis and a more limiting set of assumptions. A thorough discussion of these works is missing.\
> > > **Response**: We thank the reviewer for pointing out this prior work, which we now cite and discuss in our revision. We do cite a similar work by Ruis, et al [4] which also makes the point that intermediate supervision can help with generalisation. The main differences to our work are (1) we consider architectural modularity and the effect of implicit biases in gradient descent, rather than intermediate supervision (that is, specifying the intermediate representation directly, as in [2]); (2) in contrast to the learnability analysis in [2], we consider settings where datasets can be learned, but in different ways, either using a 'systematic' or 'non-systematic' mapping; and (3) while [2] treats the case of learning a systematic mapping from only compositional features, we investigate learning a systematic mapping with both compositional and non-compositional input features, to address the critical question of whether there is a tendency toward or against systematicity when both strategies are available. We hope that we have clarified what aspects of this work contribute to its novelty and urge the reviewer to reconsider their novelty scores.\
> > > $[1]$ Shalev-Shwartz et al., On the Sample Complexity of End-to-end Training vs. Semantic Abstraction Training, arXiv preprint arXiv:1604.06915, 2016.\
> > > $[2]$ Shalev-Shwartz et al., Failures of Gradient-Based Deep Learning, ICML 2017.\
> > > $[3]$ Wies et al., Sub-Task Decomposition Enables Learning in Sequence to Sequence Tasks, arXiv preprint arXiv:2204.02892, 2022.\
> > > $[4]$ Ruis, Laura, et al. "A benchmark for systematic generalization in grounded language understanding." Advances in neural information processing systems 33 (2020): 19861-19872.

---

> > > > ### Author Response · Authors · 2022-11-14
> > > > **Response to Official Review of Paper5201 by Reviewer 83KP (Part 4 of 4)**
> > > >
> > > > **Comment**: ...many results are given without any proof and we only get to see the final answer, e.g., the derivation of the singular value decomposition equations in App. C, or the partitions of the forbinuous norm in App. D. The authors should provide more details on how they got to these results...\
> > > > **Response**: Thank you, we have now added a proof of correctness for the SVD in Appendix D.1.
> > > >
> > > > We have also added proof sketches for the results in the main text. We thank the reviewer once again for their assistance on our work and in particular in improving the technical clarity. We believe that the paper is now clearer as a result. We look forward to engaging further and if there are any further changes which would improve the paper we would be keen to make them.

---

> ### Author Response · Authors · 2022-11-17
> **Eager to make any further changes the reviewer may have.**
>
> We once again would like to thank the reviewer for their helpful comments and assistance with the technical clarity of this work. We are aware that the discussion period ends tomorrow (18 November) and would like to make sure we address the reviewer's concerns. Thus, we would like to ask if there are any further changes which we could make that the reviewer thinks could improve the quality of our work? In summary the main changes we have already made based on the original review are:
> 1) We added the requested proof sketches for all Observations in the main body of work.
> 2) We made sure that the point of each section is clear, motivated our approaches in each section and ensured that each section leads better into the next.
> 3) We added the requested proof of correctness for our SVD into Appendix D.1.
> 4) We have incorporated the omitted citations and used them to contextualise our own work.
>
> We are eager to make any further suggested changes with the time remaining in the discussion period.

---

### Official Review · Reviewer_rTdC · 2022-10-25

**Confidence:** 4
**Correctness:** 4
**Technical Novelty And Significance:** 4
**Empirical Novelty And Significance:** 4
**Recommendation:** 8

**Clarity, Quality, Novelty And Reproducibility:**

**Quality**
Overall, the quality of the theoretical and empirical results is high. The claims made in the paper are backed up by solid evidence.

**Clarity**
The text is generally clear and figures are generally well-illustrated. As mentioned above, the authors may wish to further expand on the mapping between the concepts in the paper as described for the simple dataset of section 3 to more complex datasets; this can help clarify the significance of the work to the broader field.

**Originality**
To my knowledge, the paper is quite original. It is the first to propose a general mathematical formalization of systematicity, and may have a strong impact on how systematicity is viewed in the field.

**Strength And Weaknesses:**

**Strengths**
The paper is quite original. It provides a mathematical formalization of systematicity which to my knowledge is the first of its kind. Moreover, it demonstrates theoretically that modularity enhances the learning of systematic mappings in deep linear networks. Finally, the authors extend the results empirically to nonlinear networks, which boosts its significance. Overall, the paper appears very relevant to the field of systematic generalization.

**Weaknesses**
The theoretical results are understandably limited to deep linear networks on simple datasets. However, the authors may wish to expand on how the definition of systematicity (3.1) could be interpreted for more complex datasets beyond the one presented in section 3. Similarly, the authors may wish to further discuss what the mapping partitions in section 5 could correspond to in more complex datasets.


**Summary Of The Paper:**

This paper studies systematic generalization and the extent to which modular networks are better at learning systematic mappings relative to non-systematic mappings. Leveraging prior results on deep linear networks, the authors demonstrate that while non-modular networks learn a combined systematic and non-systematic mapping, a partitioned network can be biased towards learning a systematic mapping. Finally, the authors demonstrate that modularity enhances generalization on the CMNIST task.

**Summary Of The Review:**

Overall, the paper makes a novel contribution to the field of systematic generalization by mathematically formulating systematicity, and showing theoretically and empirically a link between modularity and systematicity. I believe this paper can be a strong contribution to the field.

---

> ### Author Response · Authors · 2022-11-14
> **Response to Official Review of Paper5201 by Reviewer rTdC**
>
> We thank the reviewer for their kind review and support of this work. We have revised our manuscript based on the reviewer's feedback on the weaknesses and have indicated these changes in green to make further discussion easier. The requested points on how our theory extends to more complex datasets and the generalisation of the mapping partitions to modular architectures has been added into the discussion (Section $8$). We once again thank the reviewer for helping improve this paper. If there are any further changes which would improve the paper we are keen to make them.

---

### Author Response · Authors · 2022-11-14
**General Response To All Reviewers**

We thank the reviewers for their comments and extremely helpful feedback. The consensus that our work is relevant and potentially impactful is gratifying and motivating. We have uploaded a revised paper addressing the noted concerns and look forward to further improving our paper during this rebuttal period. We have colour coded changes in the revision based on which review motivated the change. We hope this will make the discussion period more convenient and productive.

The main concern of reviewers 83KP and tTiF was clarity, and we agree clarity is paramount. We have made the following changes which we hope will substantially improve this aspect of the paper:

* We have revised Fig 1 to more clearly depict one intuitive instantiation of our dataset and setting. We now also label dataset parameters in this figure in an effort to ground their meaning.

* We have added Appendix A, which more extensively discusses a motivating example behind our space of datasets, including a new figure that depicts several points within the space. We also now describe simulations of generalization performance that we hope help clarify the link to systematicity.

* We have added sign-posting at the beginning and end of sections to highlight the main takeaways.

* We have included proof sketches to provide the intuitive reason behind our formal results for each Observation.

The reviewers' comments have sparked further ideas to improve clarity that we are working on, and we intend to post a further revision. We welcome further suggestions.

---

### Author Response · Authors · 2022-11-18
**Final Revision and Comment**

A final revision of the paper has been posted along with the code for all experiments (including the new experiments run in Appendix A). In this revision we focused further on improving clarity, specifically with regard to the partitioned Frobenius norms and their intuition. Specifically, we have changed how we describe them in the last paragraph of page 6, now saying "These norms provide a precise measure of the network's association between input and output blocks. For example, $\Gamma_x\Omega_y$-Norm depicts how much the network relies on the non-combinatorial input component ($\Gamma_x$) to label the combinatorial output component ($\Omega_y$)". Secondly, we have added a new figure - Figure 11 in Appendix E which depicts the partitioning and how each norm relates to the singular value decomposition used to obtain the network training dynamics. We believe these two changes now make the intuition for the norms much clearer.

In closing, we would like to thank the reviewers one more time for their helpful reviews and responses. We are very pleased with how our paper has improved due to this review process. Lastly, we find all of the reviewers' kind comments to be gratifying and inspiring. From noting the potential impact this work could have, to suggesting that our work could also be well placed in other impactful venues such as TMLR and JMLR. We are so pleased to have reviewers who share our excitement for this work. We believe this is a very promising sign that this work is of interest and applicable to a wide range of readers in the ICLR community, who will also share our excitement. A point the reviewers also seem to agree on. Thus, our final change has been to include our acknowledgement of the reviewers' impact on this work in the paper.

---

### Decision · Program_Chairs · 2023-01-20

**Decision:**

Accept: poster

**Justification For Why Not Higher Score:**

Although the authors conducted theoretical study on linear DNN, they do not extend the study to more complicate datasets and nonlinear networks.  The reviewers also raised some valuable concerns such as lacking intuitions behind the definitions and experimental results. The authors are encouraged to make the necessary changes to the best of their ability in the final camera-ready version of the paper.

**Justification For Why Not Lower Score:**

The theoretical study on systematic generalization is original and of significant importance to the community. To my knowledge, this is the first work that provides a mathematic definition of systematicity.
The claims are backed up by solid evidence and will inspire future work.

**Metareview: Summary, Strengths And Weaknesses:**

Summary:
This paper investigates how to achieve systematic generalization via neural module specialization from theoretical perspective. Build upon previous results on deep linear network, this work demonstrates that a partitioned network can be biased towards learning a systematic mapping, and modularity enhances generalization on the CMNIST task.

Strengths:
- The theoretical study on systematic generalization is original. To my knowledge, this is the first work that provides a mathematic definition of systematicity.
- The claims are backed up by solid evidence and will inspire future work.
- The results on deep linear network is extended to nonlinear network empirically.
- The topic is of significant importance to the community.

Weaknesses:
- The theoretical results are limited. It does not provide theoretical study on more complicate datasets and nonlinear networks.
- The paper lacks some intuitions behind the definitions and experimental results.
- The experimental setting is not very realistic.
- The work is not fully explained in the main body. Some details are moved to appendix due to the limitation of 9 main pages.

**Note From Pc:**

if the above contains the word "oral" or "spotlight" please see: "oral" presentation means -> notable-top-5% and "spotlight" means -> notable-top-25%. As stated in our emails, we are disassociating presentation type from AC recommendations

**Summary Of Ac-Reviewer Meeting:**

During the virtual meeting, all reviewers agreed that the paper has addressed an important problem in machine learning -- how to achieve systematic generalization via neural module specialization.

Reviewer rTdC considers the paper is clear and original and would like to keep the rating of 8.

Reviewer 83KP agreed that the revised version is more clear and would like to raise the score from 5 to 6.

Reviewer tTiF still thinks that the revised paper lacks some intuition and clarity.

Both Reviewers 83KP and tTiF think that it requires more than 9 pages to make the work fully explained and a journal paper would be a better place to publish this paper.